# Functional assessment of the "two-hit" model for neurodevelopmental defects in *Drosophila* and *X. laevis*

**Lucilla Pizzo**[1☉], **Micaela Lasser**[2☉], **Tanzeen Yusuff**[1], **Matthew Jensen**[1], **Phoebe Ingraham**[1], **Emily Huber**[1], **Mayanglambam Dhruba Singh**[1], **Connor Monahan**[2], **Janani Iyer**[1], **Inshya Desai**[1], **Siddharth Karthikeyan**[1], **Dagny J. Gould**[1], **Sneha Yennawar**[1], **Alexis T. Weiner**[1], **Vijay Kumar Pounraja**[1], **Arjun Krishnan**[3,4], **Melissa M. Rolls**[1], **Laura Anne Lowery**[5], **Santhosh Girirajan**[1,6]*

1 Department of Biochemistry and Molecular Biology, The Pennsylvania State University, University Park, PA, United States of America, 2 Department of Biology, Boston College, Chestnut Hill, MA, United States of America, 3 Department of Computational Mathematics, Science and Engineering, Michigan State University, East Lansing, MI, United States of America, 4 Department of Biochemistry and Molecular Biology, Michigan State University, East Lansing, MI, United States of America, 5 Department of Medicine, Boston University Medical Center, Boston, MA, United States of America, 6 Department of Anthropology, The Pennsylvania State University, University Park, PA, United States of America

☉ These authors contributed equally to this work.
* sxg47@psu.edu

**Data Availability Statement:** Gene expression data for the Drosophila RNAi knockdown of homologs of 16p12.1 genes and controls have been submitted to the NCBI Gene Expression Omnibus

## Abstract

We previously identified a deletion on chromosome 16p12.1 that is mostly inherited and associated with multiple neurodevelopmental outcomes, where severely affected probands carried an excess of rare pathogenic variants compared to mildly affected carrier parents. We hypothesized that the 16p12.1 deletion sensitizes the genome for disease, while "second-hits" in the genetic background modulate the phenotypic trajectory. To test this model, we examined how neurodevelopmental defects conferred by knockdown of individual 16p12.1 homologs are modulated by simultaneous knockdown of homologs of "second-hit" genes in *Drosophila melanogaster* and *Xenopus laevis*. We observed that knockdown of 16p12.1 homologs affect multiple phenotypic domains, leading to delayed developmental timing, seizure susceptibility, brain alterations, abnormal dendrite and axonal morphology, and cellular proliferation defects. Compared to genes within the 16p11.2 deletion, which has higher *de novo* occurrence, 16p12.1 homologs were less likely to interact with each other in *Drosophila* models or a human brain-specific interaction network, suggesting that interactions with "second-hit" genes may confer higher impact towards neurodevelopmental phenotypes. Assessment of 212 pairwise interactions in *Drosophila* between 16p12.1 homologs and 76 homologs of patient-specific "second-hit" genes (such as *ARID1B* and *CACNA1A*), genes within neurodevelopmental pathways (such as *PTEN* and *UBE3A*), and transcriptomic targets (such as *DSCAM* and *TRRAP*) identified genetic interactions in 63% of the tested pairs. In 11 out of 15 families, patient-specific "second-hits" enhanced or suppressed the phenotypic effects of one or many 16p12.1 homologs in 32/96 pairwise combinations tested. In fact, homologs of *SETD5* synergistically interacted with homologs of *MOSMO* in both *Drosophila* and *X. laevis*, leading to modified cellular and brain phenotypes, as well as

(GEO; https://www.ncbi.nlm.nih.gov/geo/) under accession number GSE151330, and the raw RNA Sequencing files are deposited in the SRA (Sequence Read Archive) with BioProject database (https://www.ncbi.nlm.nih.gov/bioproject/) accession number PRJNA635495. Source code for the RNA-Sequencing and network analysis is available on the Girirajan lab GitHub page at https://github.com/girirajanlab/16p12_fly_project.

**Funding:** This work was supported by the National Institutes of Health (NIH) R01GM121907 (https://www.nih.gov) and resources from the Huck Institutes of the Life Sciences (https://www.huck.psu.edu) to S.G, Fulbright Commission Uruguay-Agencia Nacional de Investigacion e Innovacion (https://fulbright.org.uy/en/) to L.P, NIH T32-GM102057 to M.J., and NIH R01GM085115 to M.M.R. The funders had no role in study design, data collection and analysis, decision to publish, or preparation of the manuscript.

**Competing interests:** The authors have declared that no competing interests exist.

axon outgrowth defects that were not observed with knockdown of either individual homolog. Our results suggest that several 16p12.1 genes sensitize the genome towards neurodevelopmental defects, and complex interactions with "second-hit" genes determine the ultimate phenotypic manifestation.

## Author summary

Copy-number variants, or deletions and duplications in the genome, are associated with multiple neurodevelopmental disorders. The developmental delay-associated 16p12.1 deletion is mostly inherited, and severely affected children carry an excess of "second-hits" variants compared to mildly affected carrier parents, suggesting that additional variants modulate the clinical manifestation. We studied this "two-hit" model using *Drosophila* and *Xenopus laevis*, and systematically tested how homologs of "second-hit" genes modulate neurodevelopmental defects observed for 16p12.1 homologs. We observed that 16p12.1 homologs independently led to multiple neurodevelopmental features and weakly interacted with each other, suggesting that interactions with "second-hit" homologs potentially have a higher impact towards neurodevelopmental defects than interactions between 16p12.1 homologs. We tested 212 pairwise interactions of 16p12.1 homologs with "second-hit" homologs and genes within conserved neurodevelopmental pathways, and observed modulation of neurodevelopmental defects caused by 16p12.1 homologs in 11 out of 15 families, and 16/32 of these changes could be attributed to genetic interactions. Interestingly, we observed that *SETD5* homologs interacted with homologs of *MOSMO*, which conferred additional neuronal phenotypes not observed with knockdown of individual homologs. We propose that the 16p12.1 deletion sensitizes the genome to multiple neurodevelopmental defects, and complex interactions with "second-hit" genes determine the clinical trajectory of the disorder.

## Introduction

Rare recurrent copy-number variants (CNVs) account for about 15% of individuals with neurodevelopmental disorders, such as autism, intellectual disability, and schizophrenia [1,2]. While certain CNVs were initially associated with specific neuropsychiatric diagnoses, such as the 16p11.2 deletion and autism [3,4], 3q29 deletion and schizophrenia [5], and 15q13.3 deletion and epilepsy [6], variable expressivity of phenotypes has been the norm rather than the exception for these CNVs [7]. A notable example of this is the 520-kbp deletion encompassing seven genes on chromosome 16p12.1, which is associated with multiple neuropsychiatric disorders, including intellectual disability/developmental delay (ID/DD), schizophrenia, and epilepsy [8,9]. Furthermore, a large-scale study on a control population reported cognitive defects in seemingly unaffected individuals with the 16p12.1 deletion [10], suggesting that the deletion is sufficient to cause neuropsychiatric features. In contrast to other pathogenic CNVs that occur mostly *de novo*, the 16p12.1 deletion is inherited in more than 95% of individuals from a mildly affected or unaffected carrier parent [8,9,11]. In fact, affected children with the deletion were more likely to carry another large CNV or deleterious mutation elsewhere in the genome ("second-hit") compared to their carrier parents [8,9], providing evidence that additional rare variants modulate the effect of the deletion. These results suggest that the 16p12.1 deletion confers significant risk for disease and sensitizes the genome for a range of neuropsychiatric

outcomes, while additional rare variants in the genetic background determine the ultimate phenotypic trajectory.

The extensive phenotypic variability and lack of chromosomal events, such as translocations and atypical deletions, have made causal gene discovery for variably-expressive CNVs such as the 16p12.1 deletion challenging. In particular, the developmental and neuronal phenotypes associated with each individual 16p12.1 gene and the interaction models that explain how "second-hit" genes modulate the associated phenotypes have not been assessed. Therefore, a systematic evaluation of developmental, neuronal, and cellular defects caused by reduced expression of individual 16p12.1 genes, as well as their interactions with each other and with "second-hit" genes from patients with the deletion, would allow us to understand the functional basis of the variable phenotypes associated with the deletion. *Drosophila melanogaster* and *Xenopus laevis* serve as excellent models for systematic evaluation of developmental and tissue-specific effects of multiple genes and their genetic interactions, as they are amenable for rapid genetic manipulation and high-throughput evaluation. In fact, *Drosophila* have been classically used to study the roles of genes and genetic interactions towards developmental and neurological phenotypes [12–14]. For example, Grossman and colleagues overexpressed human transgenes from chromosome 21 in flies and identified synergistic interactions between *DSCAM* and *COL6A2*, which potentially contribute to the heart defects observed in individuals with Down syndrome [15]. Furthermore, functional assays using *X. laevis* have uncovered developmental defects, behaviors, and molecular mechanisms for several homologs of genes associated with neurodevelopmental disorders, such as *NLGN1* [16], *CACNA1C* [17], *GRIK2* [18], and *PTEN* [19].

Using *Drosophila* and *X. laevis* models, we recently found that multiple genes within the variably expressive 16p11.2 and 3q29 deletion regions individually contribute to neurodevelopmental defects [20,21], suggesting that no single gene could be solely causative for the wide range of defects observed with deletion of an entire region. Moreover, we identified complex genetic interactions within conserved biological pathways among homologs of genes affected by these CNVs. For example, fly and *X. laevis* homologs of *NCBP2* enhanced the neuronal and cellular phenotypes of each of the other homologs of 3q29 genes [21], while fly homologs of 16p11.2 genes interacted in cellular proliferation pathways in an epistatic manner to enhance or suppress phenotypes of individual homologs [20]. In fact, several aspects of the interactions observed in our studies were also functionally or thematically validated in vertebrate model systems, providing further evidence for the utility of these models to study complex genetic interactions [22,23]. While our previous work showed pervasive interactions of homologs within regions associated with neurodevelopmental disease, the deletions within these regions occur primarily *de novo* [11], indicating a strong phenotypic impact associated with these CNVs. In contrast, the 16p12.1 deletion is mostly inherited and more frequently co-occurs with "second-hit" variants in affected individuals than other pathogenic CNVs [11], suggesting that interactions involving "second-hit" genes may confer a higher impact towards the variable neurodevelopmental phenotypes compared to those caused by interactions between genes within the CNV region.

Here, using *Drosophila melanogaster* and *X. laevis* as two complementary model systems of development, we present the first systematic assessment of conserved genes within the 16p12.1 deletion towards developmental, neuronal, and cellular phenotypes in functional models. We found that knockdown of each individual 16p12.1 homolog affects multiple phenotypic domains of neurodevelopment, leading to developmental delay and seizure susceptibility, brain size alterations, neuronal morphology abnormalities, and cellular proliferation defects. These defects were modulated by simultaneous knockdown of homologs of genes in established neurodevelopmental pathways and transcriptome targets, as well as homologs of genes

that carried "second-hits" in affected children with the deletion, through genetic interactions and "additive" effects. Our results suggest a model where reduced expression of each individual gene within 16p12.1 is sufficient to sensitize the genome towards distinct neurodevelopmental defects, which are then modulated by complex interactions with "second-hit" genes.

## Results

### Multiple homologs of 16p12.1 genes contribute to *Drosophila* and *X. laevis* development

We identified four conserved fly homologs out of the seven human protein coding 16p12.1 genes using reciprocal BLAST and orthology prediction tools (**S1 Table**) [24]. Using RNA interference (RNAi) and the *UAS-GAL4* system [25], we reduced the expression of the four fly homologs in a tissue-specific manner, and studied their individual contributions towards developmental, neuronal, and cellular defects (**Fig 1**). A complete list of the fly lines used in this study and full genotypes for all experiments are provided in **S1 File**. We authenticated the RNAi lines by confirming 40–60% expression of the four homologs using RT-qPCR (**S1 Fig**). We note that the genes are represented with fly gene names along with human counterparts at first mention in the text, and as fly genes with allele names in the figures.

We first assessed the global role of 16p12.1 homologs during development by decreasing their expression ubiquitously using the *da-GAL4* driver, and detected larval lethality with knockdown of *Sin* (*POLR3E*) and larval and pupal lethality with knockdown of *UQCR-C2* (*UQCRC2*) (**Figs 2A** and **S2**). Wing-specific *bx^MS1096^-GAL4* mediated knockdown led to severe phenotypes for *Sin* and severe defects and lethality for *UQCR-C2* fly models, recapitulating the observations made with ubiquitous knockdown (**Figs 2A** and **S2**) and suggesting a role for these homologs in signaling pathways required for early development [26–28]. Next, we evaluated whether decreased expression of the homologs leads to neuronal phenotypes frequently observed in animal models of neurodevelopmental disease, including altered lifespan, susceptibility to seizures, delayed developmental timing, changes in brain size, and dendritic arbor defects [29–34]. We observed early lethality in adult flies with nervous system-specific *Elav-GAL4*-mediated knockdown of *Sin* and *CG14182* (*MOSMO*) (**Fig 2B**), while extended lifespan was observed with knockdown of *UQCR-C2*, as previously reported for *Hsp26*, *Hsp27* [35], and *SOD* [36]. While altered mitochondrial activity has been shown to increase lifespan in *Drosophila* [37,38], further studies are necessary to understand the mechanism underlying this phenotype observed with knockdown of *UQCR-C2*. *UQCR-C2* knockdown in the nervous system also led to significantly greater recovery time when subjected to mechanical stress during bang sensitivity assays, suggesting a higher susceptibility to developing seizures [32] (**S2 Fig**). Furthermore, evaluation of developmental transitions revealed delayed pupariation and larval lethality with knockdown of *Sin*, indicating a possible role for this gene in developmental timing, as well as partial larval lethality for *CG14182* (**Fig 2C**). We also analyzed neuronal morphology in *Drosophila* class IV sensory neurons using the *ppk-Gal4* driver [31,39,40], and identified reduced complexity of dendritic arbors during development for *CG14182* (**Fig 2D**). Measurements of total area of the developing third instar larval brain led to reduced brain sizes with pan-neuronal knockdown of *CG14182* and *Sin* (**Figs 2E** and **S3**), which corresponded with a decreased number of cells in the brain lobe stained with anti-phosphorylated-Histone 3 (pH3), a marker for proliferating cells (**Fig 2F**). Interestingly, *Sin* knockdown also led to a reduction in the number of apoptotic cells, as indicated by staining with anti-Death-caspase-1 (Dcp-1) (**S3 Fig**), likely reflecting its role in both proliferation and apoptotic processes [41,42].

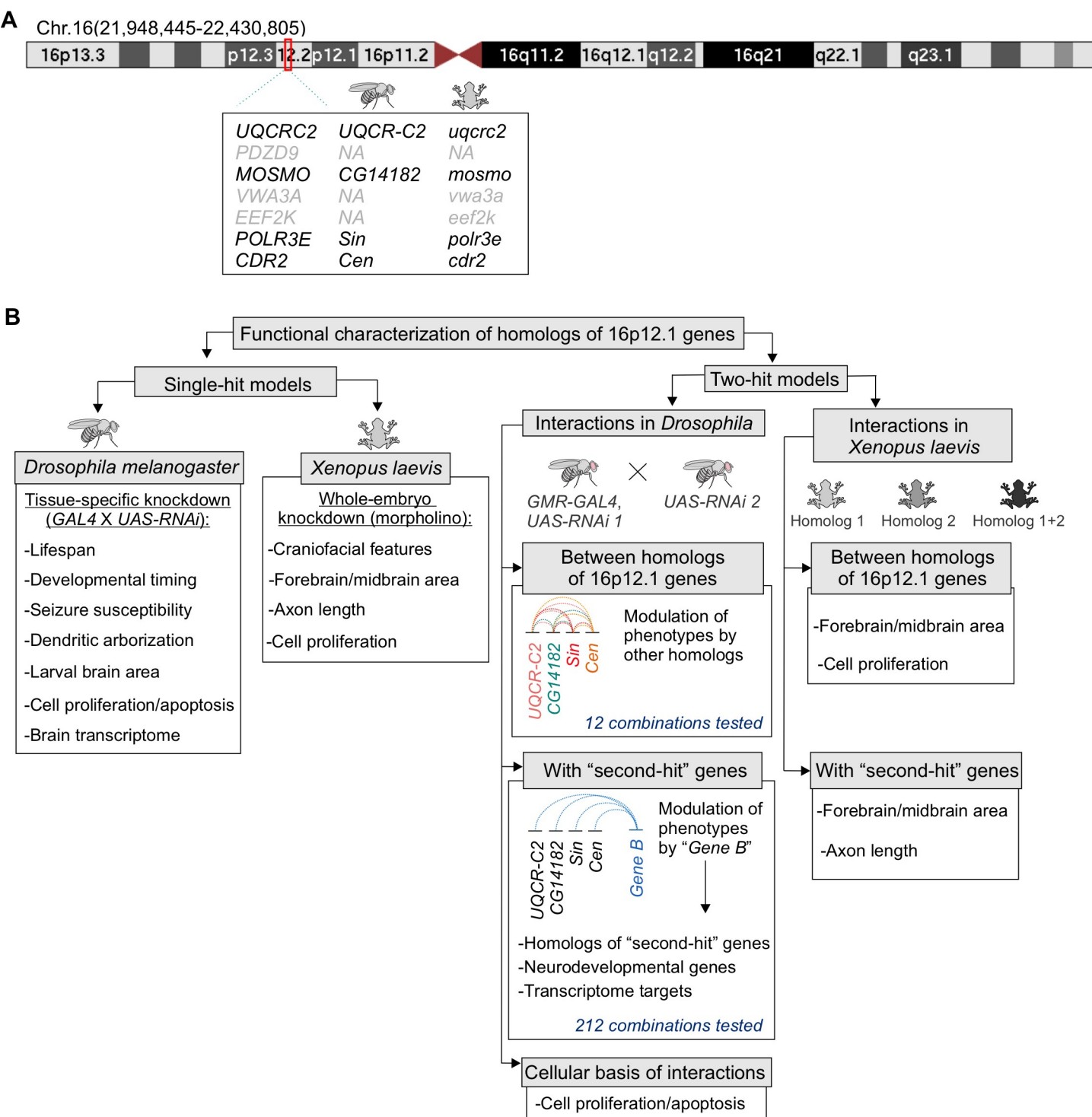

**Fig 1. Strategy to evaluate the individual contributions of homologs of 16p12.1 genes and their interactions with "second-hit" genes towards neurodevelopmental phenotypes.** (A) Ideogram of human chromosome 16 indicating the deleted region on UCSC genome build GRCh37, hg19 (chr16:21,948,445–22,430,805) (also known as 16p12.2 deletion). Seven protein coding genes are located within the 16p12.1 deletion region, including *UQCRC2*, *PDZD9*, *MOSMO*, *VWA3A*, *EEF2K*, *POLR3E*, and *CDR2*. Four out of the seven genes are conserved in both *Drosophila melanogaster* and *Xenopus laevis*. (B) We performed global and functional domain-specific phenotypic assessment using RNAi lines and tissue-specific knockdown in *Drosophila*, and morpholino-mediated whole embryo knockdown in *X. laevis*, to identify individual contributions of 16p12.1 homologs towards different developmental and neuronal features. We next evaluated the effect of pairwise knockdown of 16p12.1 homologs towards eye phenotypes in *Drosophila*, and brain size and cellular proliferation defects in *X. laevis*. We characterized 212 interactions between the 16p12.1 homologs and homologs of patient-specific "second-hit" genes identified in children with the deletion, genes within conserved neurodevelopmental pathways, and differentially-expressed genes identified from RNA-seq analysis. We found that homologs of "second-hit" genes participate in complex genetic interactions with 16p12.1 homologs to modulate neurodevelopmental and cellular phenotypes.

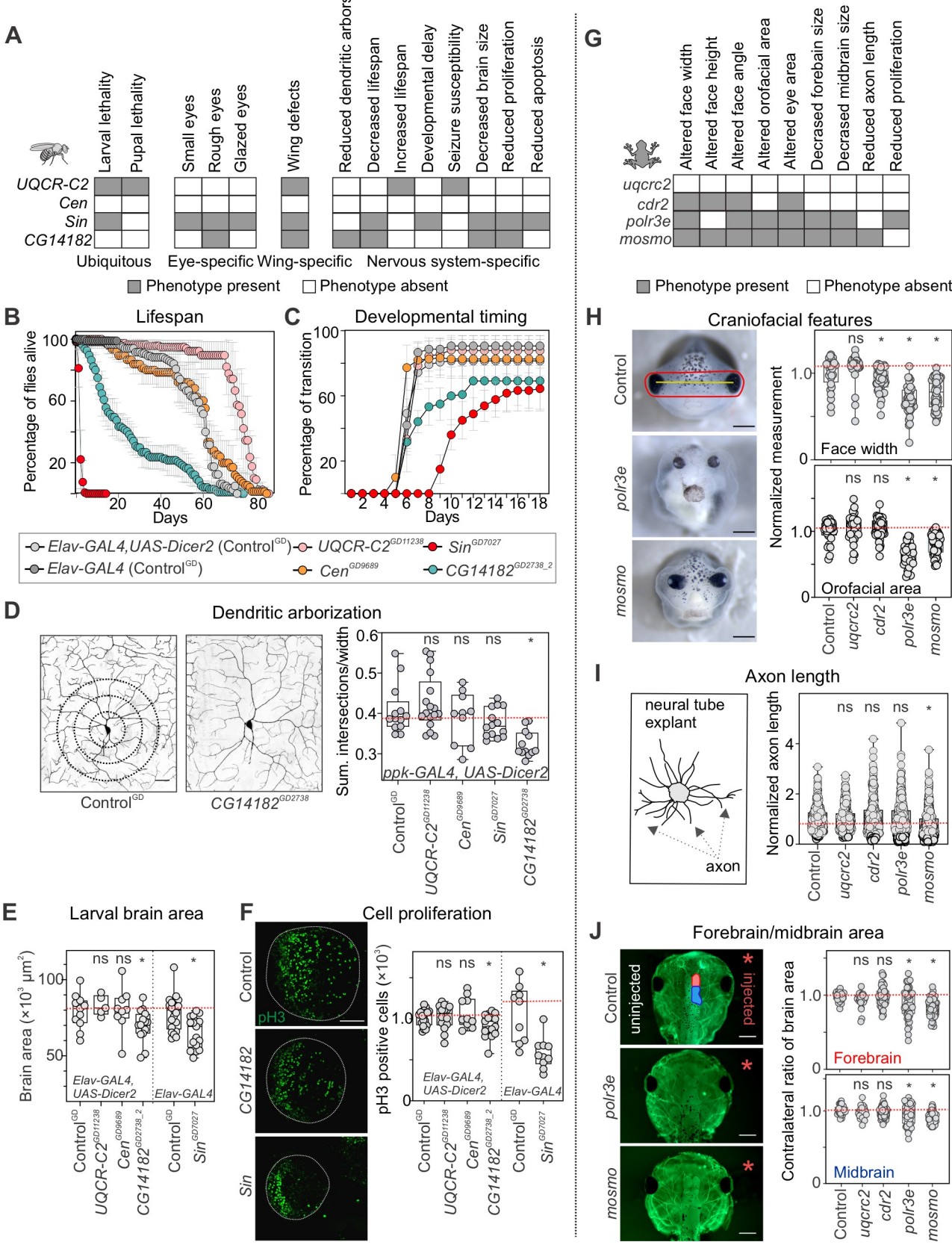

**Fig 2. Multiple homologs of 16p12.1 genes contribute to neurodevelopmental defects in *Drosophila melanogaster* and *X. laevis*.** (**A**) Schematic showing multiple phenotypes affected by tissue-specific knockdown of individual 16p12.1 homologs in *Drosophila melanogaster*. Ubiquitous knockdown was achieved with *da-GAL4*, eye-specific knockdown with *GMR-GAL4*, wing-specific knockdown with *bx^MS1096^-GAL4*, and nervous system-specific with *ppk-GAL4* or *Elav-GAL4*. See **S2A–S2C Fig** for details on phenotypes observed for individual fly lines. (**B**) Nervous-system mediated knockdown using *Elav-GAL4* with overexpression of *Dicer2* at 25°C led to reduced lifespan with knockdown of *CG14182^GD2738_2^* (n = 100, one-way repeat measures ANOVA with post-hoc pairwise t-test, days 6–61, p<0.05) and increased lifespan with knockdown of *UQCR-C2^GD11238^* (n = 120, days 51–81, p<0.05). *Elav-GAL4* mediated knockdown of *Sin^GD7027^* at RT without overexpression of *Dicer2* led to reduced lifespan of adult flies (n = 160, day 1–6, p<0.05). Data represented show mean ± standard deviation of 4–8 independent groups of 20 flies for each line tested. (**C**) Nervous-system mediated knockdown led to delayed pupariation time and larval lethality for *Sin^GD7027^* (n = 180, one-way repeat measures ANOVA with post-hoc pairwise t-test, days 6–18, p<0.05) and partial larval lethality for *CG14182^GD2738_2^* (n = 120, days 7–11, p<0.05). Data represented show mean ± standard deviation of 4–9 independent groups of 30 larvae for each line tested. (**D**) Knockdown of 16p12.1 homologs in sensory class IV dendritic arborization neurons using *ppk-GAL4* with overexpression of *Dicer2* showed reduced complexity of dendritic arbors (measured as sum of intersections normalized to width) for *CG14182^GD2738^* (n = 12, two-tailed Mann-Whitney, *p = 5.35 ×10^−5^). Scale bar represents 25 µm. (**E**) Third instar larvae with nervous system-specific knockdown of 16p12.1 homologs showed reduced brain area for *CG14182^GD2738_2^* (n = 15, two-tailed Mann-Whitney, *p = 0.047) and *Sin^GD7027^* (n = 17, *p = 0.001). (**F**) Developing third instar larvae with knockdown of *CG14182^GD2738_2^* (n = 15, two-tailed Mann-Whitney, *p = 0.026) and *Sin^GD7027^* (n = 10, *p = 9.74×10^−4^) showed reduced number of phosphorylated Histone-3 (pH3) positive cells in the brain lobe (green). Scale bar represents 50 µm. All control data for *Drosophila* represents phenotypes observed for the GD VDRC control (Control^GD^) crossed with the indicated tissue-specific *GAL4* driver. (**G**) Schematic showing the phenotypes observed with knockdown of 16p12.1 homologs in *X. laevis*. (**H**) Representative images of tadpoles injected with control morpholino, indicating facial landmarks for face width (yellow) and orofacial area (red), and tadpoles with knockdown of *polr3e* and *mosmo*. Knockdown of *cdr2* (n = 54, two-tailed student's t-test, *p = 7.75 ×10^−4^), *polr3e* (n = 37, *p = 1.97×10^−13^) and *mosmo* (n = 50, *p = 1.36×10^−11^) led to decreased face width, while knockdown of *polr3e* (*p = 3.29×10^−16^) and *mosmo* (*p = 1.47×10^−8^) led to decreased orofacial area. All measures were normalized to their respective control injected with the same morpholino amount. Scale bar represents 500 µm. (**I**) Strong knockdown of *mosmo* led to decreased axon length in neural tube explants (n = 566, two-tailed student's t-test, *p = 7.40 ×10^−12^). All measures were normalized to their respective control injected with the same morpholino amount. Representative schematic for axon length measurements is shown on the left. (**J**) Representative images show forebrain (red on control image) and midbrain (blue) areas of the side injected with morpholino (right, red asterisk), which were normalized to the uninjected side (left). Strong knockdown of *mosmo* (n = 67, two-tailed student's t-test, *p<3.07×10^−13^) and *polr3e* (n = 48, *p<7.39×10^−4^) led to decreased midbrain and forebrain area of *X. laevis* tadpoles (stained with tubulin). Scale bar represents 500 µm. In all cases, *X. laevis* data represents strong knockdown of the 16p12.1 homologs, except for *cdr2*, which showed lethality and is represented with partial knockdown. All control data for *X. laevis* represents controls injected with the highest amount of morpholino (50 ng, see **S5 Fig**). Boxplots represent all data points with median, 25th and 75th percentiles, and red dotted lines indicate the control median. Statistical details, including sample size, confidence intervals, and p-values, are provided in **S6 File**. A list of full genotypes for fly crosses used in these experiments is provided in **S1 File**.

We then performed RNA-sequencing of fly heads with pan-neuronal knockdown of the 16p12.1 homologs. Gene Ontology (GO) enrichment analysis of differentially expressed genes identified enrichments for multiple cellular, developmental, and neuronal processes (**S4 Fig** and **S2 File**). We found that each 16p12.1 homolog disrupted unique sets of genes and biological functions, as 1,386/1,870 (74%) differentially expressed genes and 28/52 (53.8%) enriched GO biological process terms were uniquely disrupted by one homolog (**S4 Fig**). Notably, we also observed this trend among the human homologs of differentially-expressed genes, with 654/994 (65.8%) uniquely differentially expressed genes and 353/428 (82.5%) GO terms uniquely disrupted by the 16p12.1 homologs, suggesting that they may act within independent pathways (**S4 Fig**). For example, knockdown of *CG14182* altered the expression of fly homologs of human genes involved in synapse assembly and transmission (*NLGN1*, *CEL*) as well as histone methyltransferase binding (*NOP56*, *CBX1*). Similarly, human homologs of genes differentially expressed with knockdown of *Sin* were involved in neuronal projection, neurotransmitter release (such as *CHRNA7*, *KCNAB2*, and multiple solute carrier transport family genes, including *SLC6A1*), and GABA pathways (such as *ADCY2*, *ADCY4*, and *ADCY7*), as well as in the development of several non-neuronal organ systems, including cardiac, kidney, lung, and muscle, further indicating the importance of *Sin* towards global development.

Next, we examined developmental phenotypes associated with decreased dosage of homologs of 16p12.1 genes in *X. laevis*, a complementary vertebrate model system (**Figs 1** and **2G**). We injected homolog-specific morpholinos at two- or four-cell stage embryos to reduce the expression of each homolog to approximately 50% (partial knockdown), and further reduced expression with higher morpholino concentrations (stronger knockdown) to increase our sensitivity to detect more specific phenotypes (**S1 Fig**, see **Materials and Methods**). Reduced

expression of *mosmo* and *polr3e* led to severe craniofacial defects in stage 42 tadpoles, as measured by specific facial landmarks, while milder defects were observed for *cdr2* (**Figs 2H** and **S5**). This suggests a role for these homologs in key developmental processes involved in craniofacial morphogenesis, such as neural crest cell formation and migration [43–48], and could potentially explain the craniofacial changes observed in more than 50% of individuals with the 16p12.1 deletion [9]. We next examined axon outgrowth phenotypes in neural tube explants from stage 20–22 injected *X. laevis* embryos, and found that stronger knockdown of *mosmo* (20 ng morpholino) led to a significant reduction in axon length (**Figs 2I** and **S5**), suggesting a potential role for the homolog in cytoskeletal signaling processes involved in axon outgrowth [49]. Furthermore, stronger knockdown of *mosmo* (20 ng morpholino) and *polr3e* (20 ng morpholino) resulted in decreased forebrain and midbrain area (**Figs 2J** and **S5**), in concordance with the brain size defects we observed in *Drosophila* models. Interestingly, partial knockdown of *mosmo* (12 ng morpholino) also led to a severe reduction in forebrain and midbrain area (**S5 Fig**). Western blot analysis for whole embryo lysates using anti-pH3 antibody as a marker for cellular proliferation showed decreased proliferation with knockdown of *polr3e*, while knockdown of *mosmo* did not lead to any overt changes (**S6 Fig**). Overall, these results suggest that homologs of 16p12.1 genes individually contribute to multiple developmental defects and affect distinct developmental, neuronal, and cellular processes in *Drosophila* and *X. laevis*.

## Weak genetic interactions and combined independent effects of 16p12.1 homologs mediate neurodevelopmental defects

Our previous studies identified several potential models for how genes within CNVs combinatorially influence neurodevelopmental phenotypes [20,21,50]. As multiple homologs of 16p12.1 genes contribute towards developmental, neuronal, and cellular phenotypes, we assessed the effect of simultaneous knockdown of pairs of 16p12.1 fly homologs. The *Drosophila* eye has been widely used to identify genetic interactions that disrupt ommatidial organization during development [51], and modifier genes for homologs of several human diseases, including Spinocerebellar ataxia type 1 [52], Huntington's disease [52], and Fragile X syndrome [53], have been studied in flies. We assessed whether eye-specific *GMR-GAL4* mediated knockdown of individual homologs led to phenotypes, and evaluated the severity of eye roughness using *Flynotyper*, a tool that quantifies the levels of ommatidial disorderliness in the adult fly eye [54]. We observed that knockdown of *Sin* led to a subtle disruption of ommatidial organization compared with controls, while no such phenotypes were observed with knockdown of other homologs (**S7 Fig**). Further reduction in expression of the 16p12.1 homologs using *GMR-GAL4* and overexpression of *Dicer2* led to more severe eye phenotypes for *CG14182* and *Sin* (**S7 Fig**). As *GMR-GAL4*-mediated knockdown of the 16p12.1 homologs only exhibited modest eye phenotypes, we used *Flynotyper* scores as a sensitive quantitative trait with a wide dynamic range to assess for combinatorial effects of 16p12.1 homologs. We therefore generated *GMR-GAL4* eye-specific recombinant lines for each homolog and crossed them with multiple RNAi lines for other 16p12.1 homologs (see **Materials and Methods, S1 File**), to test a total of 30 two-hit crosses for 12 pairwise knockdowns (**Figs 1, 3A** and **S8**).

We performed two independent analyses to interpret the combinatorial effect of 16p12.1 homologs on eye phenotypes. *First*, we assessed whether the phenotypic severity observed with knockdown of a 16p12.1 homolog is enhanced or suppressed with simultaneous knockdown of a second homolog. We observed significant changes in eye severity for four pairwise knockdowns compared with single hit recombinant lines crossed with controls, which were further validated with multiple RNAi lines (**S3 File**). For example, we observed that simultaneous knockdown of *UQCR-C2* with *Sin* or *CG14182* led to an increase in eye phenotype compared

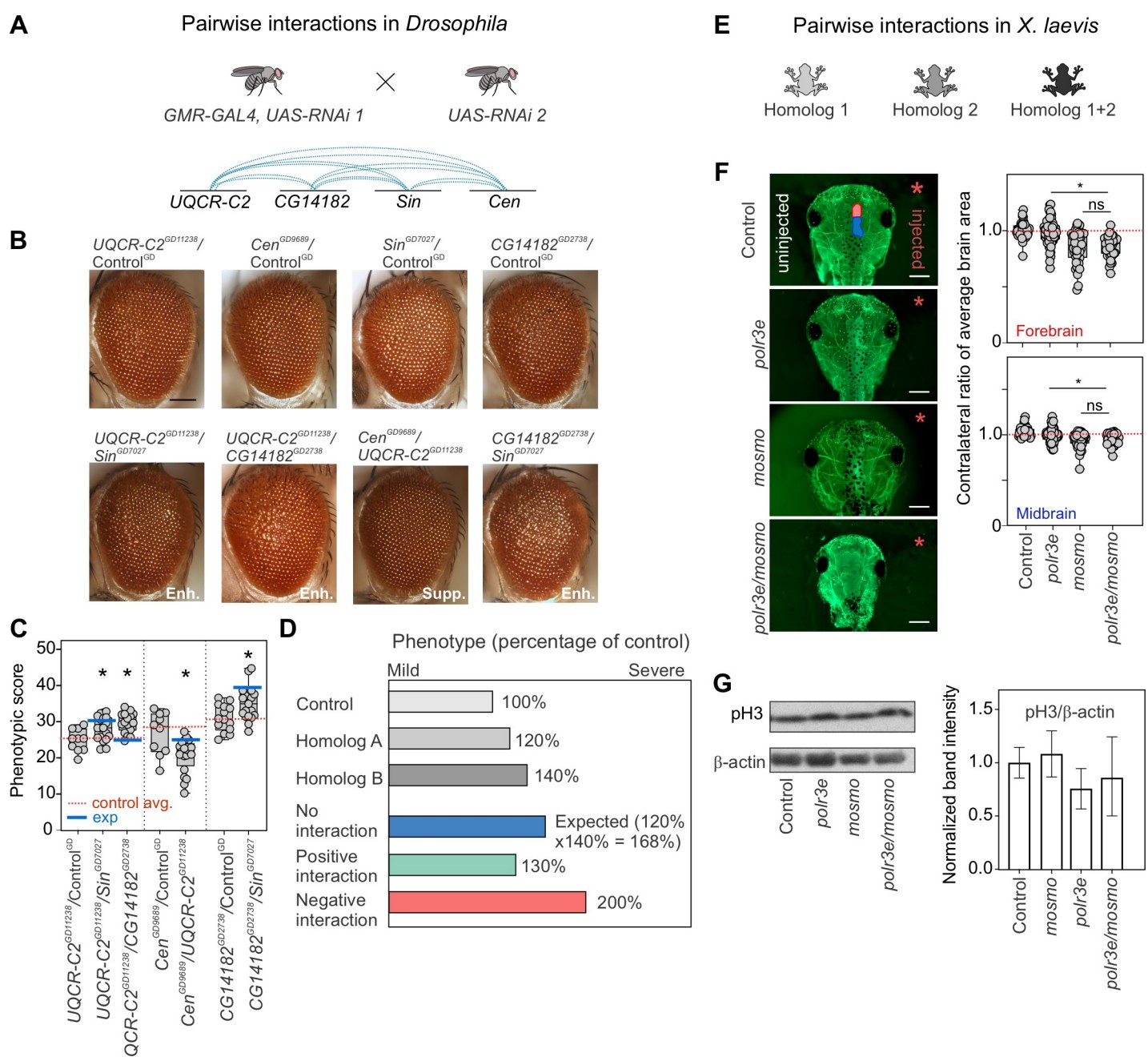

**Fig 3. Homologs of 16p12.1 genes contribute towards neurodevelopmental defects through weak genetic interactions and combined independent effects.** (**A**) We generated eye-specific *GMR-GAL4* recombinant lines for the four 16p12.1 homologs to test a total of twelve pairwise interactions for modulation of eye defects. (**B**) Representative brightfield images of *Drosophila* adult eyes for recombinant lines of 16p12.1 homologs crossed with RNAi lines for the other homologs, which show enhancement (Enh.) or suppression (Supp.) of the phenotypes observed with crosses with control. Scale bar represents 100 μm. (**C**) Simultaneous knockdown of $UQCR\text{-}C2^{GD11238}$ with $CG14182^{GD2738}$ (n = 18, two-tailed Mann-Whitney with Benjamini-Hochberg correction, *p = 0.002) or $Sin^{GD7027}$ (n = 19, *p = 0.023) led to a significant enhancement in the eye phenotype (measured using *Flynotyper* scores) compared to knockdown of $UQCR\text{-}C2^{GD11238}$ alone. Similarly, simultaneous knockdown of $CG14182^{GD2738}$ with $Sin^{GD7027}$ (n = 19, *p = 0.021) enhanced the eye phenotype observed for $CG14182^{GD2738}$ alone. Simultaneous knockdown of $Cen^{GD9689}$ with $UQCR\text{-}C2^{GD11238}$ (n = 20, *p = 0.023) led to a milder suppression of the eye phenotype compared to knockdown of $Cen^{GD9689}$ alone. Double knockdowns were compared to the recombinant lines of the 16p12.1 homologs crossed with wild-type controls for the second 16p12.1 homolog. Note that only experiments with Control^GD are represented here; see **S8 Fig** for results from other lines with KK and BL controls. (**D**) We applied a multiplicative model to identify the nature of combinatorial effects for the pairwise knockdowns tested. The expected phenotype from simultaneous knockdown of homolog A and homolog B, or when the combined effect indicates no genetic interaction (in blue), was calculated as the product of the normalized phenotypic scores (i.e. percentage of control) observed from knockdown of individual genes. Positive or alleviating genetic interactions were identified for combinations where the observed phenotype was significantly milder than expected (in green), while negative or aggravating interactions were identified when the combined phenotypes were significantly more severe than expected (in red).

One-sample Wilcoxon signed rank tests with Benjamini-Hochberg correction for multiple testing were used to identify significant interactions. (**E**) We generated double knockdowns of 16p12.1 homologs in *X. laevis* models by co-injecting embryos with morpholinos of two homologs. All double knockdown experiments were performed with partial knockdown of the genes, to avoid potential lethality with stronger knockdown. (**F**) Representative images of tadpoles stained with anti-tubulin show forebrain (red on control image) and midbrain (blue) areas of the side injected with morpholino (right, red asterisk), which were normalized to the uninjected side (left). Simultaneous knockdown of *polr3e* and *mosmo* led to decreased forebrain (n = 36, two-tailed student's t-test, *p = $1.10{\times}10^{-9}$) and midbrain area (*p = $1.98{\times}10^{-7}$), which showed no differences compared to partial knockdown of *mosmo* alone. Control data represents control injected with highest amount of morpholino (22ng). Scale bar represents 500 μm. (**G**) Representative western blots show bands for phosphorylated histone-3 (pH3) and β-actin for the uninjected control, knockdown of *polr3e*, knockdown of *mosmo*, and pairwise knockdown of *polr3e* and *mosmo* (full western blots are shown in **S6 Fig**). Bar plot shows intensity of pH3 band normalized to β-actin, with error bars representing mean ± SD. Simultaneous knockdown of *polr3e* and *mosmo* does not lead to changes in the proliferation defects observed with knockdown with *polr3e* alone. Boxplots represent all data points with median, 25th and 75th percentiles, and red dotted lines indicate the control median. Statistical details, including sample size, confidence intervals, and p-values, are provided in **S6 File**. A list of full genotypes for fly crosses used in these experiments is provided in **S1 File**.

to knockdown of *UQCR-C2* crossed with control (**Fig 3B** and **3C**). Similarly, decreased expression of *Sin* led to an enhancement of the *CG14182* eye phenotype. *Second*, to quantitatively assess whether the change in phenotypic severity due to pairwise knockdowns could be attributed to genetic interactions, we applied a "multiplicative" model to the *Flynotyper* scores (**S9 Fig**). The multiplicative model estimates the expected combined effect (i.e., no interaction) of two gene mutations as the product of the phenotypes observed with individual gene mutations (**Figs 3D** and **S9**), and identifies any deviation of the observed phenotypes from the expected values as positive (ameliorating the phenotype) or negative/synergistic (aggravating the phenotype) interactions. This strategy has been widely applied to identify fitness-based genetic interactions [55,56], and more recently, to assess for interactions contributing to non-fitness-related quantitative phenotypes, such as cell count, nuclear area [57], and protein folding in the endoplasmic reticulum [58]. After applying the multiplicative model to our pairwise interaction data, we identified five pairwise combinations of 16p12.1 homologs that were validated using multiple RNAi lines (**S2 Table**). Two pairwise combinations corresponded with no interactions, while the remaining three were positive genetic interactions, with an observed phenotype milder than expected (**S3 File**). Only one out of the four pairwise knockdowns that resulted in significant changes in eye severity compared with single hit recombinant lines crossed with control corresponded with a genetic interaction, while the rest were not validated across multiple fly lines tested (**S3 File**). To contextualize these observations, we compared the strength of genetic interactions among the 16p12.1 homologs to those of homologs of genes affected by the autism-associated 16p11.2 deletion, a region with reported pervasive genetic interactions [20]. We quantified the magnitude of genetic interactions using "interaction scores", defined as the $\log_2$ ratio between the observed and expected phenotypic values from the multiplicative model (see **Materials and Methods**), and found significantly lower interaction scores for the 16p12.1 homologs compared to the 16p11.2 homologs (**S4 File**).

We further investigated the effects of the combined knockdown of homologs of *MOSMO* and *POLR3E*, genes that individually contributed to multiple defects in both fly and *X. laevis* models, towards *X. laevis* development (**Figs 1** and **3E**). Pairwise interactions in *X. laevis* models were tested using partial knockdown of the homologs to avoid potential lethality with stronger knockdown. Partial pairwise knockdown of *polr3e* (10 ng morpholino) and *mosmo* (12 ng morpholino) showed significantly reduced forebrain and midbrain area when compared to knockdown of *polr3e* alone (**Fig 3F**), but not when compared to knockdown of *mosmo* alone. Similarly, we assessed whether *mosmo* and *polr3e* interact to modulate cellular proliferation processes during *X. laevis* development and did not observe any changes in anti-pH3 signals with combined knockdown of *polr3e* and *mosmo* compared with knockdown of *polr3e* alone (**Figs 3G** and **S6**). Overall, our analysis in *Drosophila* and *X. laevis* suggest that 16p12.1 homologs contribute towards neurodevelopmental phenotypes through both weak genetic interactions and combined independent effects.

## Homologs of 16p12.1 genes interact with genes in conserved neurodevelopmental pathways

We recently identified genetic interactions between fly homologs of CNV genes and conserved genes in neurodevelopmental pathways, providing functional evidence that phenotypes of CNV genes are modulated by key neurodevelopmental genes [20,21]. As our functional analyses showed that knockdown of each 16p12.1 homolog resulted in multiple neuronal and developmental phenotypes, we hypothesized that genes involved in conserved neurodevelopmental pathways could modulate phenotypes due to knockdown of 16p12.1 homologs through genetic interactions. We therefore performed 255 crosses to test 116 pairwise gene combinations between eye-specific recombinant lines for each of the four 16p12.1 homologs and 13 homologs of known neurodevelopmental genes and 39 homologs of transcriptional targets (**Figs 4A** and **S10**, **S2** and **S3 Tables**). As validation, we used multiple RNAi, mutant or overexpression lines when available (**Fig 1**). Details of the number of homologs, fly lines, and crosses used for all interaction experiments are provided in **S2 Table**. *First*, we screened for 55 combinations between homologs of 16p12.1 genes and 13 homologs of human genes in established developmental pathways, such as synapse function (*Prosap*/*SHANK3*), cell division (*Pten*/*PTEN*), and chromatin modulation (*kis*/*CHD8*), as well as genes functionally related to 16p12.1 homologs [54,59–61]. Using *Flynotyper* to quantify adult eye defects and the multiplicative model to identify genetic interactions, we identified interactions specific to an individual 16p12.1 homolog or those involving multiple homologs (**S11–S16 Figs** and **S3 File**). For example, *CG10465* (*KCTD13*) negatively interacted with *UQCR-C2* and *CG14182*, leading to significantly more severe phenotypes than expected using the multiplicative model (**Fig 4B**). Similarly, simultaneous knockdown of *Sin* with *kis* led to an exaggerated eye phenotype, suggesting negative interactions between the genes (**Fig 4B**). Overall, we confirmed 22 interactions out of the 55 pairwise combinations (40%) tested, including both positive (12/55) and negative (10/55) effects (**Figs 4C, 4D** and **S17** and **S2 Table**). *Next*, to identify interactors of 16p12.1 homologs towards developmental functions and pathways (**S10 Fig** and **S3 Table**), we screened for interactions of the homologs with 25 dysregulated fly genes selected from our transcriptome studies as well as 14 genes within enriched Gene Ontology categories, such as nervous system development and function (*Dscam1*, *Asp*, *mGluR*, *NaCP60E*), protein folding (*Hsp23*, *Hsp26*, *Hsp70Ab*), and muscle contraction (*Actn*, *ck*). We identified interactions for 42 out of 61 tested pairs (68.8%, **S2 Table**), validated using additional lines when available (**S3 File**). For example, knockdown of *Gat* (*SLC6A1*), *Dscam4* (*DSCAM*), *Nipped-A* (*TRRAP*), and *aurB* (*AURKB*) each modified the eye phenotype due to knockdown of *Sin* through positive or negative genetic interactions (**Figs 4C**, **S13**, **S16** and **S3 File**). Furthermore, the protein-folding gene *Hsp26* (*CRYAA)* was differentially expressed with knockdown of *Cen*, and its overexpression enhanced the phenotype of *Cen* through a negative interaction (**Figs 4C** and **S12** and **S3 File**). Overall, we identified 64 pairwise interactions between the 16p12.1 homologs and genes from established neuronal functions and transcriptome targets (**Fig 4C and 4D** and **S2 Table**), suggesting that phenotypes of 16p12.1 homologs can be modulated by genes within multiple neurodevelopmental pathways through genetic interactions.

## Homologs of patient-specific "second-hit" genes modulate phenotypes of 16p12.1 homologs

We recently found that an increased burden of rare variants (or "second-hits") outside of disease-associated CNVs, such as 16p11.2 deletion, 15q13.3 deletion, and 16p12.1 deletion, contributed to variability of cognitive and developmental phenotypes among affected children with these CNVs [8,9,11]. In fact, we found that severely affected children with the 16p12.1

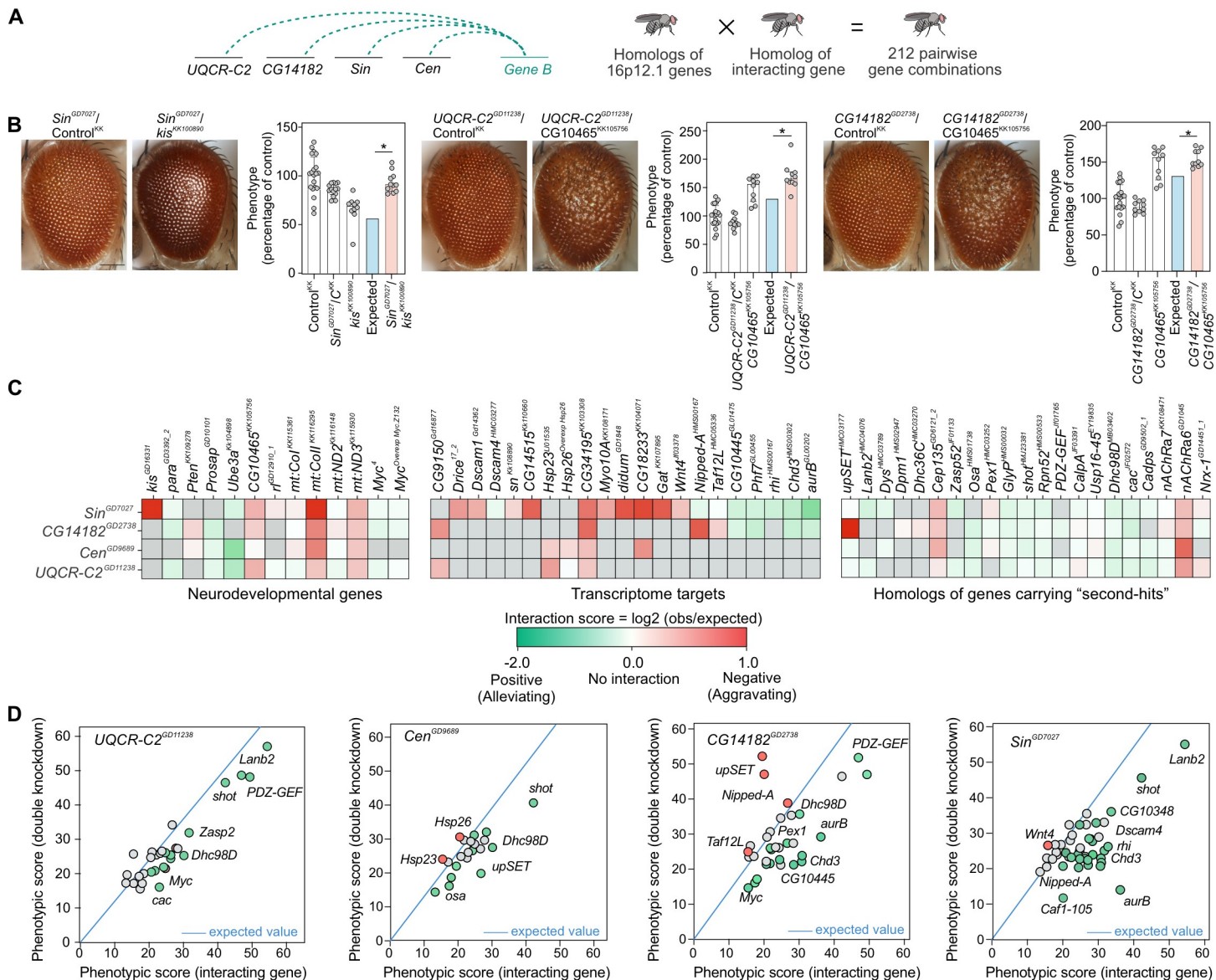

**Fig 4. Homologs of 16p12.1 genes show complex interactions with conserved neurodevelopmental genes and homologs of patient-specific "second-hit" genes. (A)** We evaluated how homologs of genes outside of the CNV region (*Gene B*), including genes carrying "second-hit" variants in children with the 16p12.1 deletion, genes within conserved neurodevelopmental pathways, and transcriptome targets, affect the phenotypes observed for homologs of 16p12.1 genes. We crossed eye-specific recombinant lines for each homolog with a total of 124 RNAi, mutant or overexpression lines for 76 interacting genes to test a total of 212 pairwise gene combinations. (**B**) Representative brightfield images of *Drosophila* adult eyes for recombinant lines of 16p12.1 homologs crossed with background-specific controls (Control[KK], also represented as C[KK]) or RNAi lines for *kis* and *CG10465*, are shown as examples of genetic interactions between the 16p12.1 homologs and homologs of neurodevelopmental genes. Bar plots show normalized phenotypes (median ± interquartile range) for 16p12.1 recombinant lines crossed with background-specific control or with RNAi lines for interacting genes. *Sin[GD7027]* negatively interacted with *kis[KK100890]* and led to a more severe phenotype (two-tail one-sample Wilcoxon signed rank test with Benjamini-Hochberg correction, n = 11, *p = 0.012, in red) than expected (in blue) under a multiplicative model. Similarly, *CG10465[KK105756]* negatively interacted with *UQCR-C2[GD11238]* (n = 10, *p = 0.024) and *CG14182[GD2738]* (n = 10, *p = 0.015), leading to more severe eye phenotypes than expected. Phenotypes are represented as percentage of average, i.e. normalized to *Flynotyper* scores from control flies carrying the same genetic background as the interacting gene. Scale bar represents 100 μm. (**C**) Heatmaps show interaction scores calculated as the log₂ ratio between the average of observed and expected phenotypic scores. Positive scores represent negative aggravating genetic interactions (in red), while negative scores represent positive alleviating interactions (in green). Grey boxes indicate pairwise crosses that were not tested or were not validated by multiple lines. A complete list of interaction scores is provided in **S4 File**. (**D**) Scatter plots depict interactions tested for 16p12.1 homologs. The plots show the average phenotypic score of the interacting gene on the x-axis, and the average observed phenotypic score for the pairwise knockdown on the y-axis. The blue line represents the expected phenotypic score of the pairwise knockdown calculated for each 16p12.1 homolog (value of first hit crossed with control, such as *UQCR-C2[GD11238]* × Control[BL]), and all possible phenotypic scores (ranging from 0 to 60) of the interacting genes are represented on the x-axis. All positive and negative (validated or potential) interactions are represented in green and red, respectively, and fly lines of genes with no significant interactions are shown in grey. Only lines from the BDSC stock center are represented here; **S17 Fig** shows scatter plots representing VDRC stock lines.

deletion had additional loss-of-function or severe missense variants within functionally intolerant genes compared to their mildly affected carrier parents [8,9,11]. We hypothesized that homologs of genes carrying patient-specific "second-hits" modulate the effects of individual 16p12.1 homologs not only "additively" but also through genetic interactions (**S10 Fig** and **S3 Table**). To test this, we performed 227 crosses to study 96 pairwise interactions between eye-specific recombinant lines for each of the four 16p12.1 homologs and 46 RNAi or mutant lines for 24 homologs of patient-specific "second-hit" genes identified in 15 families with the 16p12.1 deletion (**Figs 1** and **S18** and **S2–S4 Tables**) [9]. Out of the 96 combinations tested, we identified 32 pairwise knockdowns that modulated the phenotype of a 16p12.1 homolog, confirmed with additional lines when available, for 11 out of 15 families carrying "second-hit" genes (**Fig 5A–5C** and **S2–S4 Tables**). In fact, the phenotypic effects of 16 out of 32 combinations were attributed to genetic interactions (**Fig 4D** and **S3 File**). Interestingly, we observed that different "second-hit" homologs showed distinct patterns of interactions with homologs of 16p12.1 genes (**Fig 5B** and **S3 File**). For example, the affected child in family GL_11 carried "second-hit" pathogenic mutations in *NRXN1* and *CEP135* (**Fig 5A**). Knockdown of the fly homolog *Nrx-1* enhanced the eye phenotype caused by knockdown of *Sin* and *UQCR-C2*, while simultaneous knockdown of *Cep135* with *UQCR-C2* led to lower phenotypic score compared to knockdown of *UQCR-C2* alone (**Fig 5B and 5C**). While the two-hit phenotypes were not significantly different from the expected combined effects of the individual genes in the above cases (**S3 File**), we observed genetic interactions between *Nrx-1* and *CG14182*, and *Cep135* and *Sin* or *Cen*, suggesting potential functional connections between these genes (**Fig 5D**). Interestingly, for 11/96 combinations tested with the multiplicative model, we found that the phenotype of the pairwise knockdown was more severe compared to the phenotype observed with the knockdown of an individual 16p12.1 homolog, but significantly less severe than the expected effects, suggesting potential buffering against deleterious combined independent effects of the genes [62,63] (**Figs 4C** and **5D** and **S3 File**). In another example, the affected child in family GL_01 carried inherited "second-hit" variants in *LAMC3* and *DMD*, as well as a *de novo* loss-of-function mutation in the intellectual disability-associated and chromatin regulator gene *SETD5* [64] (**S18 Fig**). Knockdown of *Lanb2*, homolog of *LAMC3*, enhanced the phenotype caused by knockdown of *UQCR-C2*, although they positively interacted towards a milder phenotype than expected (**Fig 4D** and **S3 File**). Furthermore, *upSET*, homolog of *SETD5*, led to enhancements of the phenotypes caused by knockdown of *Sin* and *CG14182* (**S3 File**). Interestingly, while the phenotype caused by simultaneous knockdown of *Sin* and *upSET* was not different from expected using the multiplicative model, *upSET* synergistically interacted with *CG14182*, leading to an enhanced eye phenotype with pairwise knockdown (**Fig 6A**). To assess the cellular changes affected by this interaction during development, we tested for alterations in apoptosis and proliferation in the third instar larval eye discs, and found that simultaneous knockdown of *CG14182* and *upSET* led to an increased number of cells undergoing proliferation and apoptosis compared to knockdown of *CG14182* alone (**Fig 6B**). Interestingly, we also identified interactions between *CG14182* and other chromatin modifier genes, including *Nipped-A*, a transcriptional target of *Sin*, and *Osa*, homolog of the "second-hit" gene *ARID1B*, identified in family GL_13 (**S16 Fig** and **S3 File**). These interactions also modulated cellular proliferation and apoptosis processes in the developing eye discs observed with knockdown of *CG14182* (**S19 Fig**).

We further evaluated whether interactions between the fly homologs of *POLR3E* and *MOSMO* with *SETD5* were also conserved during vertebrate development, and studied brain and axon outgrowth phenotypes of homologs of these genes in *X. laevis* (**Fig 1**). We observed that simultaneous knockdown of *polr3e* and *setd5* led to smaller forebrain and midbrain areas compared with *polr3e* knockdown alone (**S20 Fig**). Similarly, simultaneous knockdown of

Interactions of 16p12.1 homologs with patient-specific "second-hits"

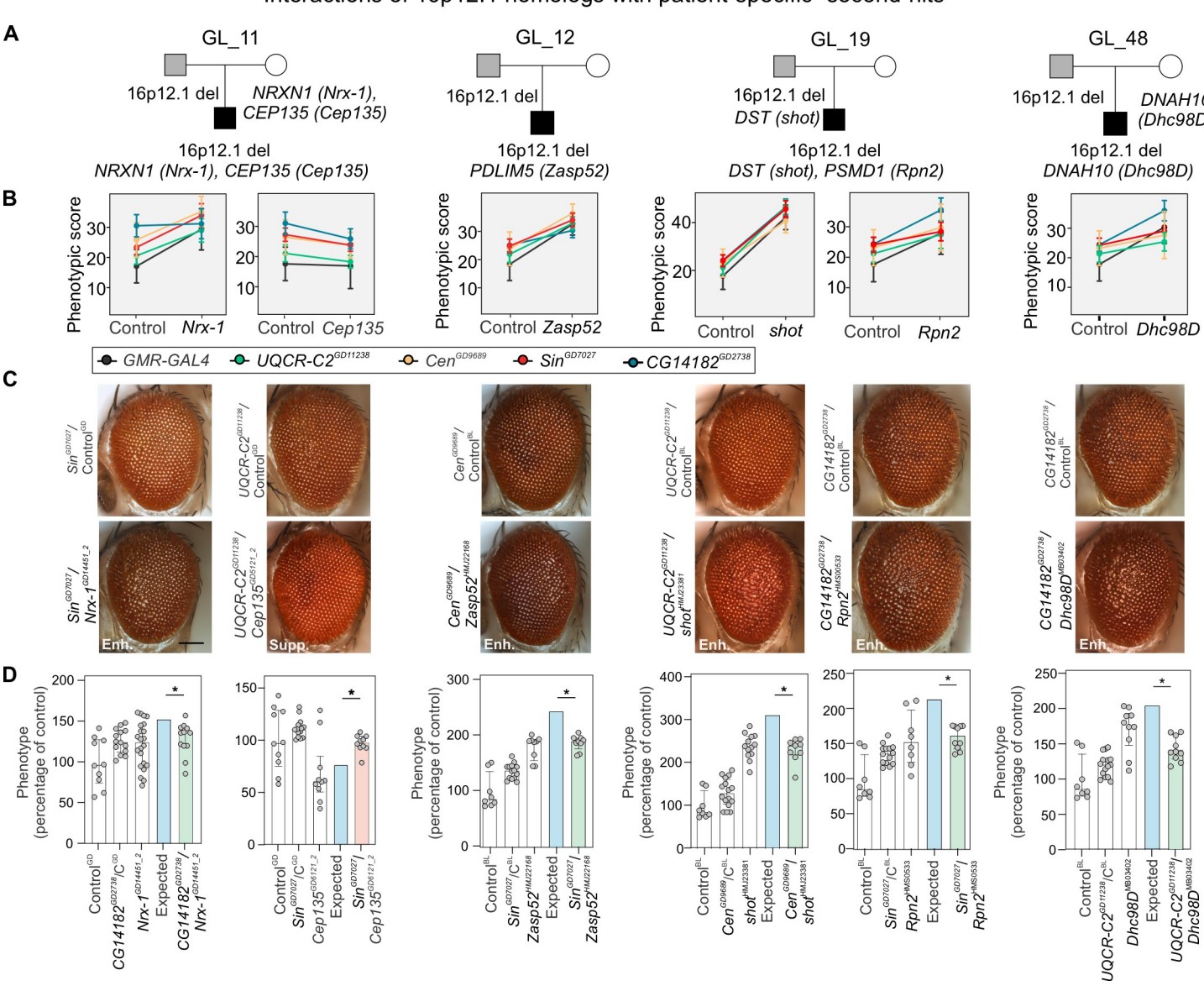

**Fig 5. Homologs of patient-specific "second-hits" modulate phenotypes of 16p12.1 homologs.** (**A**) Representative pedigrees of families with 16p12.1 deletion (affected child in black, carrier parent in grey) that were selected to study the effect of homologs (represented within parenthesis) of genes carrying "second-hits" towards phenotypes of homologs of 16p12.1 genes. (**B**) Observed phenotypic changes of 16p12.1 homologs by patient-specific "second-hit" homologs. Plots show the changes in *Flynotyper* scores (mean ± s.d.) for *GMR-GAL4* control (grey) or recombinant lines of 16p12.1 homologs crossed with either background-specific control line (left) or with "second-hit" homologs (right). We note that represented changes in *Flynotyper* scores for *Cen*$^{GD9689}$/*Nrx-1*$^{GD14451\_2}$, *UQCR-C2*$^{GD11238}$/*Zasp52*$^{HMJ22168}$, and *Sin*$^{GD7027}$/*Zasp52*$^{HMJ22168}$ were not validated with multiple RNAi lines for the "second-hit" homolog. *Flynotyper* values for all the tested pairwise knockdowns are shown in **S11**–**S14 Figs** and validated enhancements and suppressions (using Mann-Whitney tests) are shown in **S15 and S16 Figs**. (**C**) Representative brightfield adult eye images for pairwise knockdowns that enhanced (Enh.) or suppressed (Supp.) phenotypes of 16p12.1 homologs are shown. Scale bar represents 100 μm. (**D**) Examples of genetic interactions identified between the 16p12.1 homologs and homologs of patient-specific "second-hit" genes using the multiplicative model. Bar plots show normalized phenotypes (median ± interquartile range) for the 16p12.1 recombinant lines crossed with background-specific controls (Control$^{GD}$ or Control$^{BL}$, also represented as C$^{GD}$, C$^{BL}$, respectively) or with RNAi lines for *Nrx-1*$^{GD14451\_2}$, *Cep135*$^{GD6121\_2}$, *Zasp52*$^{HMJ22168}$, *shot*$^{HMJ23381}$, *Rpn2*$^{HMS0533}$, *Dhc98*$^{DMB03402}$. *Sin*$^{GD7027}$ negatively interacted with *Cep135*$^{GD6121\_2}$ and led to a more aggravating phenotype (two-tail one-sample Wilcoxon signed rank test with Benjamini-Hochberg correction, n = 10, *p = 0.012, in red) than expected (in blue) under a multiplicative model, while other examples of pairwise knockdowns with homologs "second-hit" genes shown here led to positive genetic interactions (*p<0.05, in green). Details of number of homologs, fly lines and crosses, as well as a list of full genotypes for all interaction experiments are provided in **S1 File**. Statistical details, including sample size, confidence intervals, and p-values, are provided in **S6 File**.

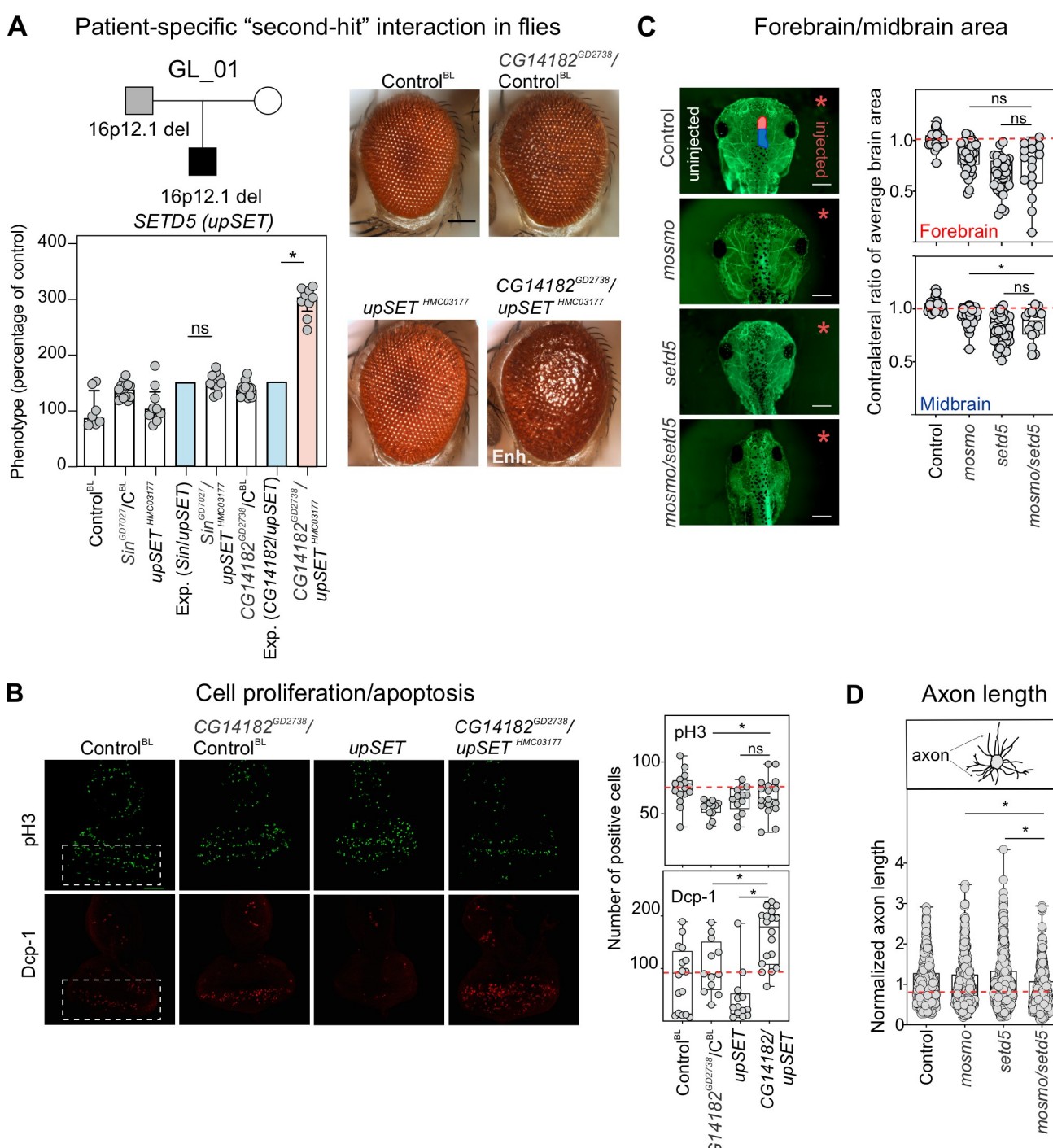

**Fig 6. Homolog of *SETD5* synergistically interacts with homolog of *MOSMO* to modify neurodevelopmental defects.** (**A**) Pedigree of a family with 16p12.1 deletion, with the proband also carrying a *de novo* pathogenic mutation in *SETD5*. Representative brightfield adult eye images for control and *GMR-GAL4* knockdown of *CG14182^GD2738^*, *upSET^HMC03177^*, and *CG14182^GD2738^*/*upSET^HMC03177^* are shown. Data show a negative genetic interaction with simultaneous knockdown of *CG14182^GD2738^* and *upSET^HMC03177^*. Bar plots show normalized phenotypes (median ± interquartile range) for recombinant lines of *CG14182^GD2738^* and *Sin^GD7027^* crossed with background-specific control (Control^BL^, also represented as C^BL^) or *upSET^HMC03177^*. An aggravating phenotype is observed with *CG14182^GD2738^*/*upSET^HMC03177^* (two-tailed one-sample Wilcoxon signed rank test with Benjamini-Hochberg correction, n = 9, *p = 0.018, in red) compared with expected (in blue). (**B**) Representative confocal images of third instar larval eye discs stained with anti-phosphorylated histone-3 (pH3, green) or anti-Dcp-1 (red), markers of cellular proliferation and apoptosis, respectively. Positive pH3 or Dcp-1 cells were quantified posterior to the morphogenetic furrow, indicated by white boxes in left panels. Double knockdown of *CG14182^GD2738^*/*upSET^HMC03177^* led to increased pH3 (n = 17, two-tailed Mann-Whitney, *p = 0.046) and Dcp-1 (n = 19, *p = 0.006) positive cells compared to knockdown of *CG14182^GD2738^*

alone. The double knockdown also led to increased Dcp-1 positive cells compared to knockdown of *upSET*^HMC03177 alone (*p = 2.19×10^5). Scale bar represents 50 μm. (**C**) Representative images of tadpoles stained with anti-tubulin show forebrain (red on control image) and midbrain (blue) areas of the side injected with morpholino (right, red asterisk), which were normalized to the uninjected side (left). Partial knockdown of *mosmo* with *setd5* led to a reduction in the midbrain area compared to the knockdown of *mosmo* alone (n = 16, two-tailed student's t-test, *p = 0.047). Control data represents control injected with highest amount of morpholino (22ng). Scale bar represents 500 μm (**D**) Normalized axon length of *X. laevis* tadpoles with simultaneous knockdown of *mosmo* and *setd5* led to a significant reduction in axon length that was not observed with partial knockdown of *mosmo* (n = 438, two-tailed student's t-test, *p = 3.34 ×10^{−6}) or *setd5* (*p = 1.86 ×10^{−9}). All measures were normalized to their respective controls injected with the same morpholino amount (See **S20 Fig**). Control data represents controls injected with highest amount of morpholino (22ng). All double knockdown experiments were performed with partial knockdown of the genes, to avoid potential lethality with stronger knockdown. Boxplots represent all data points with median, 25th and 75th percentiles, and red dotted lines indicate the control median. A list of full genotypes for fly crosses used in these experiments is provided in **S1 File**. Statistical details, including sample size, confidence intervals, and p-values, are provided in **S6 File**.

*mosmo* and *setd5* led to a further reduction in midbrain area than that observed with knockdown of *mosmo* alone (**Fig 6C**). Furthermore, analysis of axon outgrowth in developing *X. laevis* embryos showed that simultaneous knockdown of *mosmo* and *setd5* led to significantly reduced axon length compared to the individual knockdowns of either *mosmo* or *setd5*, while no changes were observed with simultaneous knockdown of *polr3e* and *setd5* (**Figs 6D** and **S20**). In fact, the axon outgrowth defect observed with simultaneous knockdown of *mosmo* and *setd5* was not observed with partial knockdown of either individual homolog. This result suggests a potential interaction between *mosmo* and *setd5* during vertebrate nervous system development. Overall, our results show that interactions with "second-hit" genes can modulate neurodevelopmental and cellular phenotypes associated with homologs of 16p12.1 genes.

## Discussion

We previously described multiple models for how genes within CNVs contribute towards neurodevelopmental phenotypes [20,21,50]. Here, we analyzed neurodevelopmental defects and cellular and molecular mechanisms due to individual and pairwise knockdown of conserved 16p12.1 homologs in *Drosophila* and *X. laevis*, and evaluated how these defects are modulated by homologs of "second-hit" genes. Our results provide multiple hypotheses for how genes within the deletion contribute to neurodevelopmental phenotypes. *First*, in line with our previous findings for homologs of genes within CNV regions [20,21], our results show that no single homolog within the 16p12.1 region is solely responsible for the observed neurodevelopmental phenotypes. In fact, we observed a global developmental role for multiple 16p12.1 homologs, as well as specific roles of each homolog towards craniofacial and brain development (**S5 Table**). This was further confirmed by interactions of 16p12.1 homologs with genes in conserved neurodevelopmental pathways. Our findings are in accordance with the core biological functions described for some of these genes. *For example*, POLR3E encodes a cofactor of the RNA polymerase III, which is involved in the transcription of small RNA, 5S ribosomal RNA, and tRNA [65], while *MOSMO* is a negative regulator of the hedgehog signaling pathway [66]. *Second*, knockdown of individual homologs sensitized both model organisms towards specific phenotypes. For example, knockdown of homologs of *MOSMO* led to neuronal morphology defects and knockdown of homologs of *POLR3E* led to brain size phenotypes that correlated with cellular proliferation defects in both model systems, while knockdown of *UQCR-C2* led to seizure susceptibility in flies. *Third*, we found that the 16p12.1 homologs were less likely to interact with each other (3 interactions out of 12 pairs tested) compared to their interactions with downstream transcriptome targets (42 interactions out of 61 pairs tested, Fisher's exact test, p = 0.0077). These results suggest reduced functional overlap among the 16p12.1 homologs, an observation supported by the distinct sets of biological functions enriched among the differentially expressed genes obtained with knockdown of each individual homolog (**S4 Fig**). Beyond the four conserved homologs evaluated in this study,

little functional information is available on the other genes in the region, including *VWA3A* and *PDZD9* as well as non-protein coding genes. Results from mouse models of *EEF2K*, which encodes a kinase associated with protein synthesis elongation, have postulated associations of this gene with synaptic plasticity [67], learning and memory [68], atherosclerosis-mediated cardiovascular disease [69], and depression [70]. Although *EEF2K* could function in concert with the tested 16p12.1 genes to contribute towards neurodevelopmental features, it showed low connectivity (29th percentile) to other 16p12.1 genes in a human brain-specific interaction network compared to the pairwise connectivity of all genes in the network [71,72] (**S5 File**). *VWA3A* showed even lower connectivity (5th percentile) to other 16p12.1 genes compared to all gene pairs. Further functional analyses that include all protein-coding and non-coding genes are necessary for a comprehensive understanding of the consequences of the entire deleted region, as these genes may also contribute towards the pathogenicity of 16p12.1 deletion.

We recently showed that additional variants or "second-hits" modulate the manifestation of developmental and cognitive phenotypes associated with disease-causing variants, including intelligence quotient and head circumference phenotypes [8,9,11]. Using the 16p12.1 deletion as a paradigm for a complex disorder, we examined how homologs of genes carrying "second-hit" variants modulate the phenotypes caused by decreased expression of individual CNV homologs. For example, homologs of *ARID1B*, *CEP135* and *CACNA1A* suppressed the eye phenotypes and interacted with one or more 16p12.1 homologs. Furthermore, we identified a negative interaction between homologs of *MOSMO* and *SETD5*, which led to novel neurodevelopmental phenotypes in both *Drosophila* and *X. laevis* compared with knockdown of either individual homolog. Interestingly, mouse embryonic stem cells lacking *Setd5* exhibited dysregulation of genes involved in hedgehog signaling [73], a key pathway recently associated with *MOSMO* function [66]. Moreover, we observed that *MOSMO* and *SETD5* are highly connected to each other in a human brain-specific interaction network compared to all genes in the genome (top 84th percentile compared to all genetic interactions with *MOSMO*), suggesting that the human genes may also be functionally related [71,72]. We further observed interactions between *CG14182* and other genes with chromatin regulating function, such as *Nipped-A* (*TRRAP*) and *osa* (*ARID1B*) (**S16 Fig**). Based on these observations, we propose that while genes carrying "second-hit" variants in combination with the deletion may "additively" contribute towards more severe phenotypes, they may also interact towards developmental phenotypes, conferring high impact towards variable defects associated with the 16p12.1 deletion (**S21 Fig**). The ultimate nature of the interactions will depend on the role of the individual CNV genes towards specific phenotypes, the identity of genes carrying "second-hits", as well as the molecular complexity associated with each phenotypic domain [50].

The high inheritance rate of the 16p12.1 deletion [8,9] suggests that while it confers risk for several phenotypes, the CNV can be transmitted through multiple generations until additional variants accumulate and cumulatively surpass the threshold for severe disease [7]. In contrast, other CNVs associated with neurodevelopmental disease, such as the autism-associated 16p11.2 deletion and the 17p11.2 deletion that causes Smith-Magenis syndrome, occur mostly *de novo* and are less likely to co-occur with another "second-hit", suggesting a higher pathogenicity on their own [11,74,75]. For example, the 16p11.2 deletion occurs *de novo* in approximately 66% of the cases, and only 8% of the affected children carry another rare large CNV, in contrast to 25% of severely affected children with 16p12.1 deletion that carry a "second-hit" large CNV [11]. When we compared experimental results from 16p12.1 homologs with those from fly homologs of 16p11.2 genes [20], we found evidence that the varying pathogenicity of the CNVs could be explained by differential connectivity and combinatorial effects of genes within each region (**Fig 7A**) [76]. For example, we previously found that 24 out of the 52 tested

pairwise knockdowns of 16p11.2 homologs led to enhancement or suppression of phenotypes, significantly modifying the effect of the individual genes [20]. In contrast, only four out of twelve tested combinations between 16p12.1 homologs led to a slight change in phenotypic severity, which in aggregate showed lower phenotypic scores than those observed for pairwise knockdown of 16p11.2 homologs (**Figs 7B** and **S22**). In fact, using a multiplicative model, we found that the magnitude of interactions between homologs of 16p11.2 genes was stronger than that observed between 16p12.1 homologs (**Fig 7C**). Moreover, transcriptome analyses showed a higher overlap of differentially expressed genes among 16p11.2 homologs compared to 16p12.1 homologs, further suggesting a higher functional relatedness among the 16p11.2 genes (**Fig 7D**). We similarly compared the connectivity of genes within both CNV regions in a human brain-specific interaction network [71,72], and found that 16p11.2 genes were more strongly connected to each other than were 16p12.1 genes, and were also more strongly connected to each other than with other genes in the genome (**Figs 7E, 7F** and **S22**). Furthermore, the connectivity of 16p12.1 genes was lower than 99.6% of connectivity values of simulated sets of six 16p11.2 genes (0.4th percentile). In fact, the connectivity values of genes within 16p12.1 and 16p11.2 were in the lower (~6th) and higher (~79th) percentiles of connectivity values from simulated sets of contiguous genes in the genome, respectively (**S23 Fig**). Interestingly, genes connecting pairs of 16p11.2 genes were enriched for genes intolerant to functional variation (**Fig 7E and 7G**), such as *ASH1L*, a histone methyltransferase activator and autism candidate gene [77], and *CAMK2B*, a protein kinase gene causative for intellectual disability [78]. In contrast, connector genes unique to 16p12.1 genes were not associated with neurodevelopmental disease or enriched for genes intolerant to variation (**S5 File**). This suggests that the 16p11.2 deletion disrupts a tight network of key genes in the brain, including other neurodevelopmental genes and genes with disease relevance [79,80]. Overall, we propose that 16p12.1 genes contribute towards multiple neurodevelopmental phenotypes through weak genetic interactions and "additive" effects and exhibit less functional connectivity compared with 16p11.2 genes, leading to a high transmissibility of the deletion and allowing for "second-hit" variants to modulate neurodevelopmental phenotypes.

Our study provides the first systematic analysis of individual and pair-wise contributions of 16p12.1 homologs towards neurodevelopmental phenotypes and associated cellular and molecular mechanisms, and identifies a key role of genetic interactions with "second-hit" homologs towards variability of phenotypes. Our work does not intend to recapitulate human disease, but rather highlights the basic cellular roles of individual conserved genes and their interactions towards neurodevelopmental phenotypes. As such, these findings should be further examined in higher-order model systems, including mouse and human cellular models. Our functional analyses suggest a model where 16p12.1 genes sensitize an individual towards defects in different domains of neurodevelopment, but the ultimate phenotypic manifestation may depend on complex interactions with "second-hits" in the genetic background.

## Materials and methods

### Ethics statement

All *X. laevis* experiments were approved by the Boston College Institutional Animal Care and Use Committee (Protocol #2016–012), and were performed according to national regulatory standards.

### *Drosophila* stocks and genetics

Using *Ensembl* database [81], NCBI Protein-Protein BLAST tool [82], and DRSC Integrative Ortholog Prediction Tool (DIOPT) [24], we identified four homologs out of the seven genes

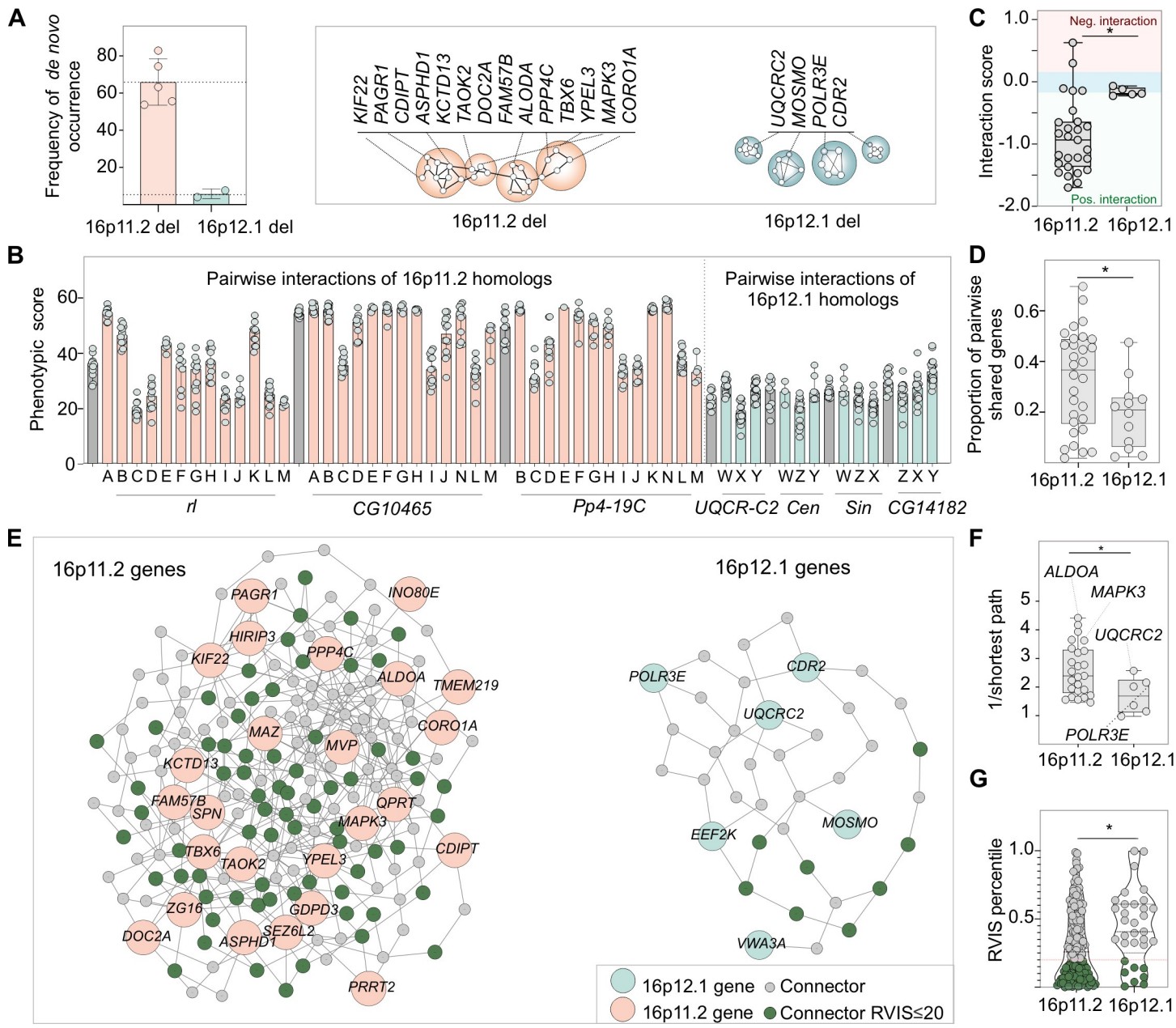

**Fig 7. Functional relatedness of genes within disease-associated CNV regions correspond with higher pathogenicity.** (**A**) Bar plot shows frequency of reported *de novo* occurrence of the 16p12.1 deletion [9,11] compared to the autism-associated 16p11.2 deletion [11,74,75]. Schematic shows a model for higher functional connectivity of genes within the 16p11.2 region compared to the 16p12.1 region. Only genes with *Drosophila* homologs are represented. (**B**) Phenotypic scores of individual 16p11.2 homologs (grey) are significantly enhanced or suppressed by a second 16p11.2 homolog (orange). In contrast, little variation in phenotypic scores is observed for 16p12.1 homologs (grey) with simultaneous knockdown of another homolog (green). The interacting homologs are labeled as follows: A: *Pp4-19C* (*PPP4C*), B: *CG17841* (*FAM57B*), C: *coro* (*CORO1A*), D: *Ald1* (*ALDOA*), E: *Rph* (*DOC2A*), F: *Tao* (*TAOK2*), G: *Asph* (*ASPHD1*), H: *klp68D* (*KIF22*), I: *Pa1* (*PAGR1*), J: *Pis* (*CDIPT*), K: *CG10465* (*KCTD13*), L: *CG15309* (*YPEL3*), M: *Doc3* (*TBX6*), N: *rl* (*MAPK3*), W: *CG14182* (*MOSMO*), X: *Cen* (*CDR2*), Y: *Sin* (*POLR3E*), Z: *UQCR-C2* (*UQCRC2*). (**C**) Pairwise knockdown of homologs of 16p11.2 genes (n = 27) show a larger magnitude of interactions compared with those among 16p12.1 homologs (n = 5, two-tailed Mann-Whitney test, *p = 0.011). Interaction values of zero (blue shade) represent no interactions, while values above or below zero represent negative (in red) and positive (in green) interactions, respectively. (**D**) Pairs of 16p11.2 homologs exhibit a higher proportion of shared differentially-expressed genes compared to pairs of 16p12.1 homologs (n = 30 for 16p11.2, n = 12 for 16p12.1, two-tailed Mann-Whitney test, *p = 0.031). (**E**) Network diagram shows connections between human 16p11.2 or 16p12.1 genes within a brain-specific interaction network. 16p12.1 genes are indicated in green, 16p11.2 genes in orange, connector genes in grey, and connector genes that are intolerant to functional variation (RVIS ≤ 20th percentile) in dark green. *C16orf92* and *C16orf54* for 16p11.2 and *PDZD9* for 16p12.1 were not present in the brain network and were therefore excluded from the network analysis. (**F**) Genes within the 16p11.2 region show higher average pairwise connectivity in a human brain-specific network, measured as the inverse of the shortest paths between two genes, compared to 16p12.1 genes (n = 25 for 16p11.2, n = 6 for 16p12.1, two-tailed Mann-Whitney, *p = 0.036, see **S5 File**). (**G**) 16p11.2 connector genes have lower RVIS percentile scores compared to 16p12.1 connector genes (n = 166 for 16p11.2, n = 33 for 16p12.1,

two-tailed Mann-Whitney, *p = 0.017, see S5 File). Functionally-intolerant genes are represented in dark green. Boxplots represent all data points with median, 25th and 75th percentiles. Statistical details are provided in S6 File.

within the 16p12.1 deletion region in *Drosophila melanogaster* (S1 Table). No fly homologs were present for three genes, including *VWA3A*, *PDZD9* and *EEF2K*. Similar strategies were used to identify fly homologs of conserved neurodevelopmental genes and genes carrying "second-hits" in children with the 16p12.1 deletion. Fly Atlas Anatomy microarray expression data from FlyBase confirmed the expression of each 16p12.1 homolog in the nervous system during *Drosophila* development (S1 Table) [83], and expression data from Xenbase [84] confirmed the expression of the homologs in *X. laevis* brain.

Multiple RNAi lines were used to test neurodevelopmental defects of 16p12.1 homologs (S1 File). RNAi, mutant, or overexpression lines for fly homologs were obtained from the Vienna Drosophila Resource Center (VDRC), Bloomington *Drosophila* Stock Center (BDSC) (NIH P40OD018537), or Kyoto Stock Center (S1 File). The following lines used were generated in various research labs: *Drice*[17–1] and *Drice*[17–2] from Bergmann lab [85], *GluRIIB*[Overexp EGFP] from Sigrist lab [86], *Hsp26*[Overexp Hsp26] from Benzer lab [35], and *Hsp70Ab*[Overexp Hsp70-9.1] and *Hsp70Ab*[Overexp Hsp70-4.3] from Robertson lab [87]. Tissue-specific knockdown of homologs of 16p12.1 genes was achieved using the *UAS-GAL4* system [25], with specific *GAL4* lines including *w*[1118];*dCad-GFP*, *GMR-GAL4/CyO* (Zhi-Chun Lai, Penn State University), *w*[1118]; *GMR-GAL4; UAS-Dicer2* (Claire Thomas, Penn State University), *w*[1118],*mcd8-GFP*, *Elav-GAL4/Fm7c;; UAS-Dicer2* (Scott Selleck, Penn State University), *w*[1118],*Elav-GAL4* (Mike Groteweil, VCU), *w*[1118];;*Elav-GAL4,UAS-Dicer2* (Scott Selleck, Penn State University), *w*[1118];*da-GAL4* (Scott Selleck, Penn State University), *w*[1118],*bx*[MS1096]-*GAL4;; UAS-Dicer2* (Zhi-Chun Lai, Penn State University), and *UAS-Dicer2; ppk-GAL4, UAS-mCD8-GFP* (Melissa Rolls, Penn State University). A list of full genotypes for all fly lines and crosses tested in this study is provided in S1 File. Fly crosses were reared on a cornmeal-sucrose-dextrose-yeast medium at room temperature (RT), 25˚C or 30˚C. For all experiments, RNAi lines were compared to a control with the same genetic background to account for background-specific effects (S1 File). Three different controls were used: *w*[1118] from VDRC (GD stock # 60000), in which inverted repeats are inserted by P-element insertion; *y,w*[1118] from VDRC (KK stock # 60100), where inverted repeats are inserted by site-specific recombination; and *{y[1] v[1]; P{y[+t7.7] = CaryP}*attP2 from BDSC (TRiP stock # 36303).

## RT-quantitative PCR for *Drosophila* RNAi knockdown of 16p12.1 homologs

Decreased expression of homologs of 16p12.1 genes in the nervous system was confirmed using reverse transcription quantitative PCR (RT-qPCR) for individual *Drosophila* RNAi lines. Decreased expression of the genes was achieved using *Elav-GAL4;;UAS-Dicer2* lines, reared at 25˚C. As nervous system-specific knockdown of *Sin* with *Elav-GAL4;;UAS-Dicer2* caused developmental lethality in all three RNAi lines tested (*Sin*[GD7027], *Sin*[KK101936], *Sin*[HMC03807]), we confirmed knockdown of *Sin* using *Elav-GAL4* without overexpression of *Dicer2* and reared at RT. We note that all experiments with nervous system-specific knockdown of *Sin* were performed under these conditions. Briefly, three biological replicates, each containing 35–40 female heads, were collected after being separated by repeated freezing in liquid nitrogen and vortex cycles. Total RNA was extracted from *Drosophila* heads using TRIzol (Invitrogen, Carlsbad, CA, USA), and cDNA was generated using qScript cDNA synthesis kit (Quantabio, Beverly, MA, USA). Quantitative RT-PCR was performed in an Applied Biosystems Fast 7500 system using SYBR Green PCR master mix (Quantabio), with *rp49* as the reference gene.

Primers were designed using NCBI Primer-BLAST [88], with primer pairs separated by an intron in the corresponding genomic DNA, if possible. **S6 Table** details the primers used to quantify the level of expression of 16p12.1 homologs. The delta-delta Ct method was used to calculate the percentage of expression compared to the control [89], and statistical significance compared to the control was determined using t-tests.

## Eye and wing imaging

Eye-specific knockdown of the 16p12.1 homologs was achieved using *GMR-GAL4* driver at 30˚C. Female progeny were collected on day 2–3 and imaged using an Olympus BX53 compound microscope with LMPLan N 20X air objective and a DP73 c-mount camera at 0.5X magnification, with a z-step size of 12.1μm (Olympus Corporation, Tokyo, Japan). Individual image slices were captured using the CellSens Dimension software (Olympus Corporation, Tokyo, Japan), and were stacked into their maximum projection using Zerene Stacker software (Zerene Systems, Richland, WA, USA). Wing phenotypes were assessed in day 2–5 female progeny from *bx^MS1096^-GAL4* lines crossed to the RNAi lines at 25˚C. Adult wings were imaged using a Zeiss Discovery V20 stereoscope (Zeiss, Thornwood, NY, USA) and a ProgRes Speed XT Core 3 camera (Jenoptik AG, Jena, Germany) with a 40X objective. Adult wing images were captured using ProgRes CapturePro v.2.8.8 software. We characterized qualitative phenotypes for between 10–20 wing images, including curly, wrinkled, shriveled, dusky or vein defects, and 10–30 eye images were assessed for rough, glazed, eye size, and necrotic patches defects. Quantitative assessment of rough adult eye phenotypes was performed using a software called *Flynotyper* [20,54], which calculates a phenotypic score for each eye image by integrating the distances and angles between neighboring ommatidia. The phenotypic scores generated by *Flynotyper* were compared between RNAi lines and their respective controls using one-tailed Mann-Whitney tests, with Benjamini-Hochberg correction for multiple tests.

## Lifespan measurement

Lifespan assessment of homologs of 16p12.1 genes was performed as previously reported [90]. Briefly, fly crosses were set up at 25˚C with *Elav-GAL4;;UAS-Dicer2* for each of the fly homologs, or *Elav-GAL4* at RT for *Sin^GD7027^*. In all cases, the newly emerged progeny were collected every day for five consecutive days, and the birth date was recorded. F1 flies were allowed to mate for 24 hours, and were separated under $CO_2$ into at least four vials, each containing 20 females. Vials were transferred every 2–3 days, and the age and number of living flies were registered. One-way repeated measures ANOVA with post-hoc pairwise t-tests were performed to identify changes in lifespan for the individual 16p12.1 homologs.

## Bang-sensitive assay

Sensitivity to mechanical stress was assessed in females with decreased expression of 16p12.1 homologs in the nervous system, using *Elav-GAL4;;UAS-Dicer2* and reared at 25˚C. *Sin* was excluded from the analysis, as adult flies with *Elav-GAL4* knockdown of the gene exhibited severe motor defects. Ten female flies from the progeny were collected on day 0–1 for ten consecutive days, and experiments were performed on day 2–3. Flies were individually separated under $CO_2$ 24 hours before the experiments, collected in culture vials containing food, and transferred to another empty culture vial the day of the experiment. Identification of bang-sensitive phenotypes was performed as previously reported [91]. Each vial was vortexed at maximum speed (Fischer Scientific) for 15 seconds, and the time for each fly to recover from paralysis was registered. Differences in bang-sensitivity compared with controls were identified using two-tailed Mann-Whitney tests.

## Assessment of delay in developmental timing

Pupariation time was assessed in third instar larvae obtained from crosses between RNAi lines and $w^{1118}$;;*Elav-GAL4*,*UAS-Dicer2* or $w^{1118}$,*Elav-GAL4* flies. Developmentally-synced larvae were obtained from apple juice plates with yeast paste, and were reared for 24 hours. Thirty newly emerged first instar larvae were transferred to culture vials, for a total of four to ten vials per RNAi line tested. The number of larvae transitioning to pupae were counted every 24 hours. Significant differences in pupariation timing compared with the control across the duration of the experiment were identified with one-way repeated measures ANOVA and post-hoc pairwise t-tests.

## Dendritic arborization experiments

Class IV sensory neuron-specific knockdown was achieved by crossing the RNAi lines to *UAS-Dicer2; ppk-GAL4* driver at 25˚C in apple juice plates. First instar larvae were collected and transferred to cornmeal-based food plates for 48 hours. Z-stack images of the dorsal side of third instar larvae were obtained using a Zeiss LSM 800 (Zeiss, Thornwood, NY, USA) confocal microscope. To perform Sholl analyses, we assessed the number of intersections of dendrite branches with four concentric circles starting from the cell body and separated by 25 μm. The total number of intersections was normalized to the width of the larval hemi-segment, and significant changes compared with control were assessed using two-tailed Mann-Whitney tests.

## Measurement of larval brain area

Larval brain area was assessed in third instar larvae obtained from crosses between the RNAi lines with *Elav-GAL4*. Crosses were set up in apple plates containing yeast paste to control for size effects generated by food availability. Fifteen first instar larvae were transferred to culture vials containing a fixed volume (8–10 mL) of cornmeal-based food. Brains were dissected from third instar larva in PBS (13mM NaCl, 0.7mM $Na_2HPO_4$, and 0.3mM $NaH_2PO_4$), fixed in 4% paraformaldehyde in PBS for 20 minutes, washed three times in PBS, and mounted in Prolong Gold antifade reagent with DAPI (Thermo Fisher Scientific, P36930). Z-stacks of *Drosophila* brains were acquired every 10μm with a 10X air objective with 1.2X magnification using an Olympus Fluoview FV1000 laser scanning confocal microscope (Olympus America, Lake Success, NY). The area of the maximum projection of the Z-stack was measured using Fiji software [92]. Differences in brain area were assessed using two-tailed Mann-Whitney tests.

## RNA sequencing and differential expression analysis in *Drosophila melanogaster*

RNA sequencing was performed for three biological replicates of RNA isolated from 35–40 *Drosophila* heads with *Elav-GAL4* mediated nervous system-specific knockdown of 16p12.1 homologs as well as controls with matching drivers and rearing temperatures. cDNA libraries were prepared with TruSeq Stranded mRNA LT Sample Prep Kit (Illumina, San Diego, CA). Single-end 100bp sequencing of the cDNA libraries was performed using Illumina HiSeq 2000 at the Pennsylvania State University Genomics Core Facility, at an average coverage of 35.1 million reads/sample. Quality control was performed using Trimmomatic [93], and raw sequencing data was aligned to the fly reference genome and transcriptome build 6.08 using TopHat2 v.2.1.1 [94]. Total read counts per gene were calculated using HTSeq-Count v.0.6.1 [95]. Differences in gene expression were identified using a generalized linear model method

in edgeR v.3.20.1 [96], with genes showing a $\log_2$-fold change $>1$ or $< -1$ and with a Benjamini-Hochberg corrected FDR$<0.05$ defined as differentially expressed. Human homologs of differentially-expressed genes in flies were identified using DIOPT v7.0. Biological pathways and processes affected by downregulation of homologs of 16p12.1 genes, defined as significant enrichments of Gene Ontology (GO) terms (p$<0.05$, Fisher's exact test with Benjamini-Hochberg multiple testing correction), were identified using PantherDB [97].

## Pairwise knockdowns in the fly eye

To study genetic interactions in the fly eye, we generated recombinant stock lines for each 16p12.1 homolog by crossing RNAi lines with eye-specific *GMR-GAL4*, as detailed in **S1 File**. Various factors including presence of balancers, chromosomal insertion of the shRNA, lethality with *Elav-GAL4*, and severity of eye phenotypes with *GMR-GAL4* were considered to select RNAi lines for generating recombinant lines. For example, for *CG14182*, we used the GD RNAi line *CG14182^{GD2738}*, which showed milder eye phenotypes, in order to test a wider range of potential interactions. We assessed genetic interactions between homologs of 16p12.1 genes with each other as well as with homologs of "second-hits" identified in children with the 16p12.1 deletion, conserved neurodevelopmental genes, and select transcriptional targets. A total of 24 homologs of genes carrying "second-hits" were selected as disease-associated genes carrying rare (ExAC frequency $\leq$1%) copy-number variants, loss-of-function (frameshift, stopgain or splicing) mutations, or *de novo* or likely-pathogenic (Phred-like CADD $\geq$25) missense mutations previously identified from exome sequencing and SNP microarrays in 15 affected children with the 16p12.1 deletion and their family members [9,98,99]. We also selected seven conserved genes strongly associated with neurodevelopmental disorders [20,54] and six genes with previously described functional associations with individual 16p12.1 genes, such as mitochondrial genes for *UQCRC2* [59] and *Myc* for *POLR3E* and *CDR2* [60,61]. We also tested interactions of the 16p12.1 homologs with 25 differentially-expressed genes (or "transcriptome targets") and 14 genes selected from enriched Gene Ontology groups identified from RNA sequencing experiments (**S3 File**). Overall, we tested 212 pairwise gene interactions including 96 interactions with homologs of "second-hit" genes, 55 with neurodevelopmental genes, and 61 with transcriptome targets, using multiple RNAi, mutant or overexpression lines per gene when available (**S2 Table**).

   *GMR-GAL4* recombinant lines for the homologs of 16p12.1 genes were crossed with RNAi or mutant lines for the interacting genes to achieve simultaneous knockdown of two genes in the eye. We also tested overexpression lines for specific genes that are functionally related to 16p12.1 homologs, including *Myc*, *Hsp23*, and *Hsp26*. Our previous assessment showed no changes in phenotypic scores for recombinant lines crossed with *UAS-GFP* compared to crosses with controls, demonstrating that the lines have adequate *GAL4* to bind to two independent *UAS-RNAi* constructs [20].

   To evaluate how simultaneous knockdown of interacting genes modulated the phenotype of 16p12.1 homologs, *Flynotyper* scores from flies with double knockdowns were compared to the scores from flies obtained from crosses between 16p12.1 recombinant lines and controls carrying the same genetic background as the interacting gene. Significant enhancements or suppressions of phenotypes of 16p12.1 homologs were identified using two-tailed Mann-Whitney tests and Benjamini-Hochberg multiple testing correction (**S3 File**).

## Analysis of genetic interactions using the multiplicative model

The nature of genetic interactions between pairs of gene knockdowns was determined using a multiplicative model, which has been widely used for evaluating genetic interactions for

quantitative phenotypes in yeast, *Drosophila*, and *E. coli* models [55–58,100]. This model tests whether the observed phenotypic effect of simultaneously knocking down two genes is different from the expected product of effects due to knockdown of individual genes. We first normalized *Flynotyper* scores for each individual line and pairwise knockdown to the *Flynotyper* scores from the background-specific controls to obtain normalized phenotypic scores as "*percentage of control.*" We then calculated the expected effects as the product of the averages of normalized scores for the two individual gene knockdowns, and compared the expected scores to the normalized scores from pairwise knockdowns using a two-tailed one-sample Wilcoxon signed ranked test with Benjamini-Hochberg correction. The distributions of the expected and observed values for all individual pairs of genes and pairwise knockdowns are shown in **S9 Fig**. Pairwise knockdowns where the observed effects were significantly higher than expected were categorized as negative, aggravating or synergistic genetic interactions, while those with observed values significantly lower than expected were considered as positive or alleviating interactions. Pairwise knockdowns where the observed effects were not significantly different from the expected effects were considered as not interacting. The magnitude of a genetic interaction was measured using "*interaction scores*", calculated as the $\log_2$ ratio between the observed and expected values (**S4 File**). Positive interaction scores indicated negative or aggravating genetic interactions, while negative interaction scores indicated positive or alleviating genetic interactions. An interaction was considered to be validated when the observed trend was reproduced by multiple fly lines when available. Interactions assessed with only one fly line were considered as "potential negative", "potential positive" or "potential no interaction" (**S3 File**). To compare interactions of 16p12.1 homologs to 16p11.2 homologs, we obtained *Flynotyper* phenotypic scores from single and pairwise *GMR-GAL4*-mediated knockdown of 16p11.2 homologs, previously described in Iyer *et al.* [20].

## Immunohistochemistry of the developing brain and eye discs in *Drosophila melanogaster*

Third instar larvae brain or eye discs were dissected in PBS and fixed in 4% paraformaldehyde in PBT (0.3% Triton X-100 in PBS), followed by three washes with PBT. Preparations were blocked for one hour in blocking buffer (5% FBS or 1% BSA in 0.3% PBT), followed by incubation overnight at 4°C with the primary antibody. We assessed for markers of proliferation using mouse anti-pH3 (S10) (1:100; 9706, Cell Signaling Technology, Danvers, MA, USA) and apoptosis using rabbit anti-Dcp-1 (Asp216) (1:100, 9578, Cell Signaling). Secondary antibody incubation was performed using Alexa fluor 647 goat anti-mouse (1:100, Invitrogen, Carlsbad, CA, USA) and Alexa fluor 568 goat anti-rabbit (1:100, Invitrogen) for 2 hours at 25°C, followed by three washes with PBT. Tissues were mounted in Prolong Gold antifade reagent with DAPI (Thermo Fisher Scientific, P36930) prior to imaging. Z-stacks of brain lobe or eye discs were acquired every 4μm with a 40X air objective with 1.2X magnification using an Olympus Fluoview FV1000 laser scanning confocal microscope (Olympus America, Lake Success, NY). Image analysis was performed using Fiji [92]. The number of cells undergoing proliferation or apoptosis were quantified throughout the brain lobe, or posterior to the morphogenetic furrow in the developing eye discs. The total number of Dcp-1 positive cells in larval brain and eye discs, as well as pH3 cells in the eye discs, were manually counted from the maximum projections. The total number of pH3 positive cells in the larval brain were quantified using the MaxEntropy automated thresholding algorithm per slice, followed by counting the number of particles larger than 1.5 μm. Differences in the number of positive pH3 or Dcp-1 cells were compared with appropriate controls using two-tailed Mann-Whitney tests.

### *Xenopus laevis* embryos

Eggs collected from female *X. laevis* frogs were fertilized *in vitro*, dejellied, and cultured following standard methods [16,101]. Embryos received injections of exogenous mRNAs or antisense oligonucleotide strategies at the two- or four-cell stage, using four total injections performed in 0.1X MMR media containing 5% Ficoll. Embryos were staged according to Nieuwkoop and Faber [102].

### *X. laevis* gene knockdown and rescue

Morpholinos (MOs) were targeted to early splice sites of *X. laevis mosmo*, *polr3e*, *uqcrc2*, *cdr2*, and *setd5*, or standard control MO (**S7 Table**), purchased from Gene Tools (Philomath, OR). In knockdown experiments, all MOs were injected at either the 2-cell or 4-cell stage, with embryos receiving injections two or four times total. *mosmo* and control MOs were injected at 12 ng/embryo for partial and 20 ng/embryo for stronger knockdown; *polr3e* and control MOs were injected at 10 ng/embryo for partial and 20 ng/embryo for stronger; *uqcrc2* and control MOs were injected at 35 ng/embryo for partial and 50 ng/embryo for stronger; *cdr2* and control MOs were injected at 10 ng/embryo for partial and 20ng for stronger knockdown; and *setd5* and control MOs were injected at 10 ng/embryo for partial knockdown. All double knockdown experiments were performed with partial knockdown to avoid potential lethality.

Splice site MOs were validated using RT-PCR. Total RNA was extracted using Trizol reagent, followed by chloroform extraction and ethanol precipitation from 2-day old embryos injected with increasing concentrations of MO targeted to each 16p12.1 homolog, respectively. cDNA was synthesized using SuperScript II Reverse Transcriptase (Invitrogen, Carlsbad, CA, USA). PCR was performed in a Mastercycler using HotStarTaq DNA Polymerase (Qiagen, Germantown, MD, USA) following manufacturer instructions. PCR was performed using primers with sequences detailed in **S6 Table**. RT-PCR was performed in triplicate, and band intensity was measured using the densitometry function in ImageJ [103] and normalized to the uninjected control mean relative to the housekeeping control *odc1*. Phenotypes were rescued with exogenous mRNAs co-injected with their corresponding MO strategies. *X. laevis* ORF for *mosmo* was purchased from the European *Xenopus* Resource Center (EXRC, Portsmouth, UK) and gateway-cloned into pCSF107mT-GATEWAY-3'GFP destination vector. Constructs used were *mosmo*-GFP, and GFP in pCS2+. In rescue experiments, MOs of the same amount used as for the knockdown of each homolog were co-injected along with mRNA (1000pg/embryo for *mosmo*-GFP) in the same injection solution.

### Quantifying craniofacial shape and size of *X. laevis* embryos

The protocol for quantifying craniofacial shape and size was adapted from Kennedy and Dickinson [104]. Embryos at stage 42 were fixed overnight in 4% paraformaldehyde in PBS. A razor blade was used to make a cut bisecting the gut to isolate the head. Isolated heads were mounted in small holes in a clay-lined dish containing PBS. Frontal and lateral view images were taken using a Zeiss AxioCam MRc attached to a Zeiss SteREO Discovery.V8 light microscope (Zeiss, Thornwood, NY, USA). ImageJ software [103] was used to perform craniofacial measurements, including: 1) facial width, which is the distance between the eyes; 2) face height, which is the distance between the top of the eyes and the top of the cement gland at the midline; 3) dorsal mouth angle, which is the angle created by drawing lines from the center of one eye, to the dorsal midline of the mouth, to the center of the opposite eye; and 4) midface area, which is the area measured from the top of the eyes to the cement gland encircling the edges of both eyes. For all facial measurements, two-tailed student's t-tests were performed

between knockdown embryos and control MO-injected embryos with the same amount of morpholino.

## Neural tube explants, imaging, and analysis

Embryos were injected with either control MO or 16p12.1 homolog-specific MO at the 2–4 cell stage, and culturing of *Xenopus* embryonic neural tube explants from stage 20–22 embryos were performed as previously described [101]. For axon outgrowth analysis, phase contrast images of axons were collected on a Zeiss Axio Observer inverted motorized microscope with a Zeiss 20X/0.5 Plan Apo phase objective (Zeiss, Thornwood, NY, USA). Raw images were analyzed by manually tracing the length of individual axons using the NeuronJ plug-in in ImageJ [105]. All experiments were performed on multiple independent occasions to ensure reproducibility. Axon outgrowth data were normalized to controls from the same experiment to account for day-to-day fluctuations. Statistical differences were identified between knockdown embryos and control MO-injected embryos with same amounts of morpholino using two-tailed student's t-tests.

## Immunostaining for brain morphology, imaging, and analysis

For brain morphology analysis, half embryo KDs were performed at the two-cell stage. *X. laevis* embryos were unilaterally injected two times with either control MO or 16p12.1 homolog-specific MO and a GFP mRNA construct (300pg/embryo). The other blastomere was left uninjected. Embryos were raised in 0.1X MMR through neurulation, and then sorted based on left/right fluorescence. Stage 47 embryos were fixed in 4% paraformaldehyde diluted in PBS for one hour, rinsed in PBS, and gutted to reduce autofluorescence. Embryos were processed for immunoreactivity by incubating in 3% bovine serum albumin and 1% Triton-X 100 in PBS for two hours, and then incubated in anti-acetylated tubulin (1:700, T7451SigmaAldrich, St. Louis MO, USA) and goat anti-mouse Alexa Fluor 488 conjugate secondary antibody (1:1000, Invitrogen, Carlsbad, CA, USA). Embryos were rinsed in 1% Tween-20 in PBS and imaged in PBS. Removal of the skin dorsal to the brain was performed if the brain was not clearly visible due to pigment.

Images were taken at 3.2X magnification using a Zeiss AxioCam MRc attached to a Zeiss SteREO Discovery.V8 light microscope (Zeiss, Thornwood, NY, USA). Images were processed in ImageJ [103]. The areas of the forebrain and midbrain were determined from raw images using the polygon area function in ImageJ. Brain sizes were quantified by taking the ratio of forebrain and midbrain areas between the injected side versus the uninjected side for each sample. All experiments were performed on at least three independent occasions to ensure reproducibility, and data shown represent findings from multiple replicates. Statistical differences were identified between knockdown embryos and control MO injected embryos with the same amount of morpholino using two-tailed student's t-tests.

## Western blot for cell proliferation

Embryos at stage 20 to 22 were lysed in buffer (50mM Tris pH 7.5, 1% NP40, 150mM NaCl, 1mM PMSF, 0.5 mM EDTA), supplemented with cOmplete™ Mini EDTA-free Protease Inhibitor Cocktail (Sigma-Aldrich) and PhosSTOP Phosphatase Inhibitor Cocktail (Sigma-Aldrich). Blotting was carried out using rabbit polyclonal antibody to Phospho-Histone H3 (Ser10) (1:500, PA5-17869, Invitrogen), with mouse anti-beta actin (1:2500, ab8224, Abcam, Cambridge, MA, USA) as a loading control. Bands were detected by chemiluminescence using Amersham ECL Western blot reagent (GE Healthcare Bio-Sciences, Pittsburgh, PA). Band intensities were quantified by densitometry in ImageJ and normalized to the control mean relative to β-actin.

## Connectivity of 16p12.1 and 16p11.2 deletion genes in a human brain-specific interaction network

We examined the connectivity of human 16p12.1 and 16p11.2 deletion genes in the context of a human brain-specific gene interaction network that was previously built using a Bayesian classifier trained on gene co-expression datasets [71,72]. As the classifier assigned weighted probabilities for interactions between all possible pairs of genes in the genome, we first built a network that only contained the top 0.5% of all pairwise interactions (predicted weights >2.0). Within this network, we identified the shortest paths between 25/27 genes in the 16p11.2 deletion or 6/7 genes in the 16p12.1 region that were present in the network and all other protein-coding genes in the genome, using the inverse of the probabilities as weights for each interaction. For each shortest path, we calculated the overall length as a measure of connectivity between the two genes, and also identified the connector genes located within the shortest path (**S5 File**). All network analyses were performed using the NetworkX v.2.4 Python package [106]. We compared the average connectivity of the six 16p12.1 genes to the 25 16p11.2 genes, as well as the predicted pathogenicity of connector genes for 16p11.2 and 16p12.1 interactions, using two-tailed Mann-Whitney tests. To account for a larger number of genes in the 16p11.2 deletion, we also compared the median connectivity of 16p12.1 genes to the distribution of median connectivity values for 1000 simulations of six random genes from the 16p11.2 region. We further compared the median connectivity of 16p12.1 genes to the connectivity of 1000 random sets of six genes in the network. Finally, to compare these values to random sets of contiguous genes in the genome, we calculated the median connectivity among 1000 randomly-selected contiguous blocks of six and 25 genes, excluding blocks containing genes within canonical CNV regions [11]. For each comparison with simulated gene sets, we calculated one-tailed z-scores and percentile values for the observed median connectivity values (**S6 File**).

## Statistical analyses

All statistical analyses of functional data were performed using R v.3.4.2 (R Foundation for Statistical Computing, Vienna, Austria). Details for each statistical test, including sample size, p-values with and without multiple testing correction, confidence intervals, test statistics, and ANOVA degrees of freedom, are provided in **S6 File**.

## Supporting information

**S1 Fig. Expression levels of 16p12.1 homologs in *Drosophila* and *X. laevis*. (A)** *Drosophila* homologs of 16p12.1 genes were knocked down using nervous system-specific *Elav-GAL4* driver with overexpression of *Dicer2* at 25˚C. RT-qPCR confirmed 40–60% knockdown of the 16p12.1 homologs (two-tailed student's t-test, *p<0.05). As knockdown of *Sin* caused embryonic lethality in these conditions, all experiments in the nervous system and RT-qPCR were performed without overexpression of *Dicer2* and reared at room temperature (RT). *Sin*[KK101936] and *Sin*[HMC03807] were also embryonic lethal without *Dicer2*. *Sin*[GD7027–2] did not show knockdown of the homolog, and the RNAi line was therefore not used for further experiments. All experiments were performed in comparison to appropriate background-specific controls. Only one control is shown per gene. GD VDRC control is shown in all cases for simplification (Control[GD]). A list of full genotypes for fly crosses used in these experiments is provided in **S1 File**. (**B**) Normalized band intensity of RT-PCR of *X. laevis* tadpoles injected with different morpholino dosages of the 16p12.1 homologs compared to the uninjected control. Different morpholino sequences were used for the L and S alleles for *uqcrc2* and *mosmo*, while unique

sequences were used for both L and S alleles for *cdr2* or *setd5*. As the S allele has not been annotated for *polr3e*, only the L allele was targeted. Colored bars represent the dosages of morpholinos used, with grey bars indicating amounts for "partial knockdown" (approximately 50% of expression) and black bars indicating amounts for "stronger knockdown". 10 ng of morpholino was used for partial and stronger KD experiments for *mosmo* S allele, as increasing concentrations did not lead to differences in knockdown of the allele. Bar plots represent mean +/- SD, and red dotted lines indicate 50% expression. Statistical details are provided in **S6 File**.
(PDF)

**S2 Fig. Global neurodevelopmental defects with knockdown of 16p12.1 *Drosophila* homologs.** (**A**) Summary of phenotypes observed with tissue-specific knockdown of each of the 16p12.1 homologs. (**B**) Ubiquitous (*Da-GAL4*) and nervous system-specific (*Elav-GAL4*) knockdown of multiple RNAi lines for each 16p12.1 homolog showed a range of lethality and developmental defects. (**C**) Representative brightfield images of adult wings with knockdown of 16p12.1 homologs using the wing-specific driver $bx^{MS1096}$-*GAL4*. Severe phenotypes were observed for *UQCR-C2* and *Sin*, with some RNAi lines, including $UQCR-C2^{GD11238\_2}$, $UQCR-C2^{KK108812}$ and $Sin^{GD7027}$, showing lethality. Scale bar represents 500μm. (**D**) Bang sensitivity assay for adult flies with nervous system-specific knockdown of the 16p12.1 homologs showed increased recovery time for $UQCR-C2^{GD11238}$ (n = 95, two-tailed Mann-Whitney test, $^*$p = 0.003). $Sin^{GD7027}$ adult flies exhibited severe motor defects and could not be tested for the phenotype. Boxplots represent all data points with median, 25th and 75th percentiles. Statistical details are provided in **S6 File**. A list of full genotypes for fly crosses used in these experiments is provided in **S1 File**.
(PDF)

**S3 Fig. $Sin^{GD7027}$ and $CG14182^{GD2738\_2}$ lead to decreased brain area and decreased number of proliferating cells in the brain.** (**A**) Representative confocal brightfield images of nervous system-specific knockdown of 16p12.1 homologs show decreased total brain area for $Sin^{GD7027}$ and $CG14182^{GD2738\_2}$. Scale bars represent 100μm. (**B**) *Elav-GAL4* mediated knockdown led to decreased number of phosphorylated histone-3 positive cells (pH3, green) in the brain lobe (DAPI, blue) with knockdown of $Sin^{GD7027}$ (n = 10, two-tailed Mann-Whitney, $^*$p = 9.74×10$^{-4}$) and $CG14182^{GD2738\_2}$ (n = 15, $^*$p = 0.026), indicating decreased proliferation with knockdown of the homologs. Knockdown of $Sin^{GD7027}$ led to decreased number of Dcp-1 positive cells ($^*$p = 2.78×10$^{-4}$, red) in the brain lobe. Scale bar represents 50μm. Boxplots represent all data points with median, 25th and 75th percentiles. Statistical details, including sample size, confidence intervals, and p-values, are provided in **S6 File**. A list of full genotypes for fly crosses used in these experiments is provided in **S1 File**.
(PDF)

**S4 Fig. Enriched GO terms observed with knockdown of 16p12.1 fly homologs in the nervous system.** (**A**) Clusters of enriched Gene Ontology (GO) Biological Process terms for differentially expressed fly genes observed with nervous system-specific knockdown of 16p12.1 homologs (left) and their human homologs (right). While some clusters of terms overlap among 16p12.1 homologs, genes dysregulated with knockdown of individual homologs exhibit unique enrichments for GO terms, suggesting their independent action towards neuronal development. Venn diagrams show overlaps of enriched GO Complete Biological Processes terms for (**B**) differentially-expressed fly genes observed with knockdown of individual 16p12.1 homologs, or (**C**) human homologs of the fly genes. We also observed that most of the (**D**) fly homologs or (**E**) human counterparts of the differentially-expressed genes were unique

to each 16p12.1 homolog. A list of differentially expressed genes with knockdown of 16p12.1 homologs, as well as a list of all enriched GO terms for these gene sets, is detailed in **S2 File**. Venn diagrams were constructed using Venny 2.1 software (https://bioinfogp.cnb.csic.es/tools/venny).

(PDF)

**S5 Fig. Decreased dosage of 16p12.1 homologs leads to multiple neurodevelopmental phenotypes in *X. laevis*. (A)** Representative images of tadpoles injected with control morpholino or morpholinos for 16p12.1 homologs, indicating facial landmarks for face width (yellow), height (blue), angle (green), and orofacial (red) and eye (orange) area. Boxplots showing face height, width, angle, and orofacial and eye area of each knockdown compared to its own control. Knockdown of *mosmo* (n = 50, two-tailed student's t-test, $^*$p = 0.010) and *cdr2* (n = 54, $^*$p = 3.68 ×10$^{-6}$) led to increased face height. Knockdown of *cdr2* ($^*$p = 7.75 ×10$^{-4}$), *polr3e* (n = 37, $^*$p = 1.97 ×10$^{-13}$) and *mosmo* ($^*$p = 1.36 ×10$^{-11}$) led to decreased face width, while knockdown of *cdr2* ($^*$p = 1.03×10$^{-8}$), *polr3e* ($^*$p = 2.73×10$^{-4}$) and *mosmo* ($^*$p = 3.50×10$^{-7}$) led to decreased face angle. Knockdown of *polr3e* ($^*$p = 3.29 ×10$^{-16}$) and *mosmo* ($^*$p = 1.47 ×10$^{-8}$) led to decreased orofacial area, and knockdown of *polr3e* ($^*$p = 1.01×10$^{-18}$), *mosmo* ($^*$p = 7.23×10$^{-10}$) and *cdr2* ($^*$p = 0.009) led to decreased eye area. Data represents strong knockdown of the 16p12.1 homologs, except for *cdr2*, which showed lethality and is shown for partial knockdown. All measures were normalized to their respective control injected with the same morpholino amount. Scale bars represent 500μm. **(B)** Boxplots showing axon length of each knockdown compared to its own control. Strong knockdown of *mosmo* led to decreased axon length in neural tube explants (n = 566, two-tailed student's t-test, $^*$p = 7.40 ×10$^{-12}$), which was rescued by co-injection with overexpressed (OE) mRNA of the gene (n = 249, $^*$p = 4.06×10$^{-5}$). All measures were normalized to their respective control injected with the same morpholino amount. **(C)** Representative images stained with anti-tubulin show forebrain (red on control image) and midbrain (blue) areas of the side injected with morpholino (right, red asterisk), which were normalized to the uninjected side (left). Partial knockdown of *mosmo* led to decreased forebrain (n = 47, two tailed student's t-test, $^*$p = 1.18×10$^{-9}$) and midbrain ($^*$p = 1.45×10$^{-7}$) area. Graphs represent contralateral ratio of brain area compared to uninjected side of the embryo. Scale bars represent 500μm. All boxplots represent all data points with median, 25th and 75th percentiles. In each case, measurements for each knockdown were compared to controls injected with equal amounts of morpholino. Statistical details, including sample size, confidence intervals, and p-values, are provided in **S6 File**.

(PDF)

**S6 Fig. Whole western blot for phosphorylated histone-3 in *X. laevis* embryos with knockdown of *polr3e*, *mosmo* and *setd5*. (A)** Three replicate western blot experiments were performed. The intensity of bands at 17 kDa, corresponding with pH3 (top, indicated with arrow), were normalized to the β-actin loading control (bottom). **(B)** Partial knockdown of *polr3e* shows reduced band intensity with anti-pH3 antibody compared to β-actin loading control. Bar plot represents mean ± SD.

(PDF)

**S7 Fig. Knockdown of *Sin* and *CG14182* lead to disruption of the fly eye in a dosage sensitive manner. (A)** Representative brightfield adult eye images and *Flynotyper* phenotypic scores of eye-specific knockdown of 16p12.1 homologs with *GMR-GAL4* and no overexpression of *Dicer2* at 30˚C. A mild eye phenotype was observed with knockdown of *Sin*, replicated across multiple RNAi lines (two-tailed Mann-Whitney with Benjamini-Hochberg correction, $^*$p<0.05). **(B)** Representative images and *Flynotyper* scores of eye-specific knockdown of

16p12.1 homologs with *GMR-GAL4* and overexpression of *Dicer2* at 30˚C. Severe eye phenotypes were observed for all tested RNAi lines of *Sin* (*p< 1.13×10$^{-4}$) and *CG14182* (*p< 4.70×10$^{-4}$). Scale bar represents 100 μm. Boxplots represent all data points with median, 25th and 75th percentiles, and red dotted lines indicate the control median. Statistical details, including sample size, confidence intervals, and p-values, are provided in **S6 File**. A list of full genotypes for fly crosses used in these experiments is provided in **S1 File**.
(PDF)

**S8 Fig. Pairwise knockdown of homologs of 16p12.1 genes lead to moderate changes in eye severity compared to individual knockdown of genes.** (**A-D**) Pairwise knockdown of homologs of 16p12.1 genes led to subtle changes in phenotypic scores, with only four significant combinations (compared to control lines) validated by multiple RNAi lines. *Sin* and *CG14182* enhanced the eye phenotype of *UQCR-C2$^{GD11238}$*, while *Sin* enhanced *CG14182$^{GD2738}$* eye phenotype and *UQCR-C2* suppressed *Cen$^{GD9689}$* eye phenotype (two-tailed Mann-Whitney with Benjamini-Hochberg correction, *p<0.05). Boxplots represent all data points with median, 25th and 75th percentiles. Red dotted lines indicate the median of recombinant lines crossed with control. Statistical details, including sample size, confidence intervals, and p-values, are provided in **S6 File**. A list of full genotypes for fly crosses used in these experiments is provided in **S1 File**.
(PDF)

**S9 Fig. Range of observed and expected phenotypic scores using a multiplicative model.** Histograms representing the distribution of observed (in grey) and expected (in blue) phenotypic scores of *GMR-GAL4*-mediated pairwise knockdowns of 16p12.1 homologs and interacting genes (top panel) and *GMR-GAL4*-mediated single knockdowns of potential interacting genes tested (bottom panel). The distribution shows an overlap of the observed and expected phenotypic scores values.
(PDF)

**S10 Fig. Expected genetic interactions for each category of interacting gene analyzed in this study.** The diagram shows representative gene networks of each of the 16p12.1 homologs analyzed. *UQCR-C2* is shown as an example, where the gene is closely connected to transcriptome targets and functionally related genes, while neurodevelopmental genes and patient-specific "second-hit" genes distribute more randomly between gene categories that can lead to positive and negative genetic interactions (top bubble) or combined independent effects (bottom bubble). These hypotheses aligned with our results, as we identified genetic interactions for 42/61 pairwise combinations (68.8%) with direct transcriptome targets, compared to 22/55 (40%) interactions identified with functionally related and neurodevelopmental genes (p = 0.0027, Fisher's exact test). Moreover, we observed that homologs of genes carrying "second-hits" in severely affected children with the 16p12.1 deletion interacted with 16p12.1 homologs in 37/96 (38.5%) of the pairs, although the proportion of genetic interactions was not as high compared to those identified with functionally related genes or genes in neurodevelopmental pathways and transcriptome targets (64/101, 63%, Fisher's exact test, p = 0.019).
(PDF)

**S11 Fig. Phenotypic scores of the tested pairwise knockdowns of *UQCR-C2$^{GD11238}$* with homologs of "second-hit" or neurodevelopmental genes in *Drosophila* eye.** *Flynotyper* phenotypic scores of *UQCR-C2$^{GD11238}$* crossed with RNAi, mutant or overexpression lines of (**A**) neurodevelopmental genes or genes functionally related with *UQCR-C2* function, (**B and C**) homologs of "second-hits" identified in children with 16p12.1 deletion, and (**D**) transcriptome targets and functionally related groups identified in RNA-sequencing of *UQCR-C2*

knockdown model. Boxplots represent all data points with median, 25th and 75th percentiles. Red dotted lines indicate the median of recombinant lines crossed with control. A list of full genotypes and statistics, including sample size, confidence intervals, and p-values, for these experiments are provided in **S1** and **S6 Files**.
(PDF)

**S12 Fig. Phenotypic scores of the tested pairwise knockdowns of *Cen*$^{GD9689}$ with homologs of "second-hit" or neurodevelopmental genes in *Drosophila* eye.** *Flynotyper* phenotypic scores of *Cen*$^{GD9689}$ crossed with RNAi, mutant or overexpression lines of (**A**) neurodevelopmental genes or genes functionally related with *Cen* function, (**B and C**) homologs of "second-hits" identified in children with 16p12.1 deletion, and (**D**) transcriptome targets and functionally related groups identified in RNA-sequencing of *Cen* knockdown model. Boxplots represent all data points with median, 25th and 75th percentiles. Red dotted lines indicate the median of recombinant lines crossed with control. A list of full genotypes and statistics, including sample size, confidence intervals, and p-values, for these experiments are provided in **S1** and **S6 Files**.
(PDF)

**S13 Fig. Phenotypic scores of the tested pairwise knockdowns of *Sin*$^{GD7027}$ with homologs of "second-hit" or neurodevelopmental genes in *Drosophila* eye.** *Flynotyper* phenotypic scores of *Sin*$^{GD7027}$ crossed with RNAi, mutant or overexpression lines of (**A**) neurodevelopmental genes or genes functionally related with *Sin* function, (**B and C**) homologs of "second-hits" identified in children with 16p12.1 deletion, and (**D and E**) transcriptome targets and functionally related groups identified in RNA-sequencing of *Sin* knockdown model. Boxplots represent all data points with median, 25th and 75th percentiles. Red dotted lines indicate the median of recombinant lines crossed with control. A list of full genotypes and statistics, including sample size, confidence intervals, and p-values, for these experiments are provided in **S1** and **S6 Files**.
(PDF)

**S14 Fig. Phenotypic scores of the tested pairwise knockdowns of *CG14182*$^{GD2738}$ with homologs of "second-hit" or neurodevelopmental genes in *Drosophila* eye.** *Flynotyper* phenotypic scores of *CG14182*$^{GD2738}$ crossed with RNAi, mutant or overexpression lines of (**A**) neurodevelopmental genes or genes functionally related with *CG14182* function, (**B and C**) homologs of "second-hits" identified in children with 16p12.1 deletion, and (**D**) transcriptome targets and functionally related groups identified in RNA-sequencing of *CG14182* knockdown model. Boxplots represent all data points with median, 25th and 75th percentiles. Red dotted lines indicate the median of recombinant lines crossed with control. A list of full genotypes and statistics, including sample size, confidence intervals, and p-values, for these experiments are provided in **S1** and **S6 Files**.
(PDF)

**S15 Fig. Multiple homologs of patient-specific "second-hits" and neurodevelopmental genes modulate phenotypes caused by knockdown of *UQCR-C2*$^{GD11238}$ and *Cen*$^{GD9689}$.** Representative brightfield adult eye images and phenotypic scores of eyes from RNAi, mutant or overexpression lines of neurodevelopmental genes, homologs of genes with "second-hits" in children with 16p12.1 deletion, and transcriptome targets that significantly enhanced (Enh.) or suppressed (Supp.) the phenotypes of recombinant lines of *UQCR-C2*$^{GD11238}$ (**A**) or *Cen*$^{GD9689}$ (**B**) (*p<0.05, two-tailed Mann-Whitney tests with Benjamini-Hochberg correction). Scale bar represents 100 μm. Boxplots represent all data points with median, 25th and 75th percentiles. Red dotted lines indicate the median of recombinant lines crossed with control. A list of full genotypes and statistics, including sample size, confidence intervals, and p-

values, for these experiments are provided in **S1** and **S6 Files**.
(PDF)

**S16 Fig. Multiple homologs of patient-specific "second-hits" and neurodevelopmental genes modulate phenotypes caused by knockdown of *CG14182^{GD2738}* and *Sin^{GD7027}***. Representative brightfield adult eye images and phenotypic scores of eyes from RNAi, mutant or overexpression lines of neurodevelopmental genes, homologs of genes with "second-hits" in children with 16p12.1 deletion, and transcriptome targets that significantly enhanced (Enh.) or suppressed (Supp.) the phenotypes of recombinant lines of *CG14182^{GD2738}* (**A**) or *Sin^{GD7027}* (**B**) (*p<0.05, two-tailed Mann-Whitney tests with Benjamini-Hochberg correction). Scale bar represents 100 μm. Boxplots represent all data points with median, 25th and 75th percentiles. Red dotted lines indicate the median of recombinant lines crossed with control. A list of full genotypes and statistics, including sample size, confidence intervals, and p-values, for these experiments are provided in **S1** and **S6 Files**.
(PDF)

**S17 Fig. Genetic interactions between 16p12.1 homologs and neurodevelopmental genes, transcriptome targets and homologs of patient-specific "second-hit" genes.** Scatter plots depict the interactions tested for *GMR-GAL4* recombinant lines for *UQCR-C2^{GD11238}* (**A**), *Cen^{GD9689}* (**B**), *CG14182^{GD2738}* (**C**), and *Sin^{GD7027}* (**D**). The plots show the average phenotypic score of the interacting gene on the x-axis using VDRC GD (top) or KK (bottom) fly lines and the average observed phenotypic score for the double knockdown on the y-axis. Blue line represents the expected phenotypic score of the pairwise knockdown calculated as the product of the first hit phenotype (*Flynotyper* score of first hit crossed with control, such as *UQCR-C2^{GD11238}* X Control^{GD}) for each theoretical phenotypic value of interacting gene (ranging from 0 to 60) represented on x-axis. All positive and negative (validated or potential) interactions are represented in green and red, respectively, and fly lines of genes with no significant interactions are shown in grey.
(PDF)

**S18 Fig. "Second-hit" genes identified in probands with 16p12.1 deletion.** Pedigrees of 15 families with the 16p12.1 deletion, highlighting 23 genes with rare secondary likely-pathogenic mutations (CNVs and SNVs) that were identified in severely affected children with the 16p12.1 deletion. These 23 genes carrying "second-hit" mutations were selected for *Drosophila* experiments to test how their decreased expression affects the neurodevelopmental phenotypes observed for 16p12.1 homologs. Family members who carry either the 16p12.1 deletion or individual genes with "second-hits" are indicated in the pedigrees. Phenotypes observed for affected children and other family members are indicated below each pedigree.
(PDF)

**S19 Fig. *Osa* and *Nipped-A* interact with *CG14182^{GD2738}* through cellular processes during eye development. (A)** Representative brightfield adult eye images show that simultaneous knockdown of *CG14182^{GD2738}* with *osa^{IF01207}* leads to a suppressed (Supp.) eye phenotype, while simultaneous knockdown of *Nipped-A^{HMS00167}* leads to synergistic enhancement (Enh.) in eye phenotype compared to single knockdown of *CG14182^{GD2738}*. Scale bar represents 100 μm. (**B**) Representative confocal images of third instar larval eye discs stained with DAPI (blue) and anti-phosphorylated histone-3 (pH3, green) or anti-Dcp-1 (red), markers of cellular proliferation and apoptosis, respectively. Positive pH3 or Dcp-1 cells were quantified posterior to the morphogenetic furrow (indicated by white boxes). Simultaneous knockdown of *CG14182^{GD2738}* with *osa^{IF01207}* led to an increase in the number of pH3 positive cells (n = 20, two-tailed Mann-Whitney, *p = $2.33 \times 10^{-6}$) compared to the single knockdown of

$CG14182^{GD2738}$. (**C**) Simultaneous knockdown of $CG14182^{GD2738}$ with $Nipped\text{-}A^{HMS00167}$ led to a significant reduction in the number of Dcp-1 positive cells compared to single knockdown of $CG14182^{GD2738}$ (n = 15, in red, two-tailed, Mann-Whitney, $^*p = 2.54\times10^{-5}$). Scale bars represent 50 μm. Boxplots represent all data points with median, 25th and 75th percentiles. Statistical details, including sample size, confidence intervals, and p-values, are provided in **S6 File**. A list of full genotypes for fly crosses used in these experiments is provided in **S1 File**. (PDF)

**S20 Fig. *setd5* modifies phenotypes observed with knockdown of *polr3e* and *mosmo* in *X. laevis*.** (**A**) Representative images stained with anti-tubulin show forebrain (red on control image) and midbrain (blue) areas of the side injected with morpholino (right, red asterisk), normalized to the uninjected side (left). Simultaneous knockdown of *polr3e* and *setd5* in *X. laevis* led to decreased forebrain (n = 28, two-tailed student's t-test, $^*p = 6.01\times10^{-7}$) and midbrain area ($^*p = 1.67\times10^{-7}$) compared to knockdown of *polr3e* alone, which were not different to the partial knockdown of *setd5* alone (p>0.05). Scale bar represents 500 μm. (**B**) Normalized axon length of *X. laevis* tadpoles with simultaneous knockdown of *mosmo* and *setd5* showed decreased axon length different from the control injected with 22ng of morpholino (n = 438, two-tailed, student's t-test, $^*p = 2.95\times10^{-7}$) and from individual knockdown of *setd5* ($^*p = 1.86\times10^{-9}$) or *mosmo* ($^*p = 3.34\times10^{-6}$), showing a synergistic effect of decreased dosage of the homologs towards neuronal phenotypes. (**C**) Normalized axon length of *X. laevis* tadpoles with simultaneous knockdown of *polr3e* and *setd5* showed no change in axon length (two-tailed student's t-test, p>0.05). In each case, the individual knockdown was normalized and compared to the control injected with the same amount of morpholino. Boxplots represent all data points with median, 25th and 75th percentiles. Statistical details, including sample size, confidence intervals, and p-values, are provided in **S6 File**. (PDF)

**S21 Fig. Interactions between 16p12.1 homologs and "second-hits" are of higher magnitude than those between 16p12.1 homologs.** (**A**) Absolute values of scores of interactions identified between 16p12.1 homologs are lower compared to those identified between the 16p12.1 homologs and "second-hit" homologs (n = 3 for 16p12.1, n = 37 for interactions with "second-hits", two-tailed Mann-Whitney, $^*p = 0.0032$). (**B**) Comparison of absolute values of interaction scores between pairs of 16p12.1 homologs to those observed between 16p12.1 homologs and homologs of "second-hits" split by proband. Only one RNAi line per interacting gene is shown. (PDF)

**S22 Fig. 16p11.2 genes exhibit higher phenotypic scores than 16p12.1 genes and are more connected to each other than to other genes in the genome.** (**A**) Pairwise eye-specific knockdown of 16p11.2 homologs (*rl*, *CG10465* and *Pp4-19C* crossed with other 16p11.2 homologs) lead to more severe phenotypic scores compared to pairwise knockdown of 16p12.1 homologs (n = 39 for 16p11.2, n = 16 for 16p12.1, one-tailed Mann-Whitney, $^*p = 3.51 \times10^{-5}$). Grey circles represent pairwise knockdowns, while 16p11.2 and 16p12.1 single-homolog knockdowns are represented in orange and green, respectively. (**B**) Analysis of a human brain-specific network shows higher average pairwise connectivity, measured as the inverse of the shortest path between two genes, between pairs of 16p11.2 genes compared to the connectivity of 16p11.2 genes to the rest of the genome (n = 25 two-tailed Mann-Whitney, $^*p = 6.64\times10^{-3}$). This trend was not observed for 16p12.1 genes (n = 6, p>0.05). Boxplots represent all data points with median, 25th and 75th percentiles. Statistical details are provided in **S6 File**. (PDF)

**S23 Fig. 16p12.1 genes are less connected than 16p11.2 genes in a human brain-specific interaction network.** (**A**) Cumulative frequency plot shows low connectivity of 16p12.1 genes (observed value shown as red dotted line) in a human brain-specific interaction network when compared with 1000 permutations of six 16p11.2 genes (blue, $0.4^{th}$ percentile, p = 0.117, one-tailed z-score test) and random sets of genes in the genome (green, $9.2^{nd}$ percentile, p = 0.143, one-tailed z-score test). Cumulative frequency plots show 1000 simulations of bins of 25 contiguous genes (green) (**B**) or six contiguous genes (blue) (**C**), compared with the median connectivity values for 16p11.2 genes (green dotted line, $78.6^{th}$ percentile, one-tailed z-score test, p = 0.246) and 16p12.1 genes (blue dotted line, $5.7^{th}$ percentile, p = 0.130).
(PDF)

**S1 Table. *Drosophila* and *X. laevis* homologs of 16p12.1 genes.** DIOPT [24] and reciprocal BLAST [82] searches were used to identify fly homologs of 16p12.1 genes. The expression of homologs in the larval central nervous system during development was assessed using FlyAtlas Anatomy microarray expression data from FlyBase [83].
(PDF)

**S2 Table. Number of experiments and genetic interactions identified in this study.** This table lists the number of crosses, RNAi/mutant/overexpression lines, tested pairwise combination of genes, and genes that significantly enhanced or suppressed the phenotype of 16p12.1 homologs (**S2A Table**). The number of confirmed and potential negative and positive genetic interactions towards eye phenotypes using the multiplicative model is also provided (**S2B Table**).
(PDF)

**S3 Table. Hypotheses, experimental design, and conclusions obtained for experiments performed in our study.** Explanation of hypotheses, experimental design, results, and conclusions that can be drawn or hypotheses that can be further tested for each experiment performed in this study is provided.
(PDF)

**S4 Table. "Second-hit" variants identified in families with 16p12.1 deletion and tested for interactions with 16p12.1 homologs.** Genes carrying "second-hits" were previously identified [9] through exome sequencing and SNP microarrays in 15 children with the 16p12.1 deletion, and selected as disease-associated genes carrying rare (ExAC frequency ≤1%) copy-number variants, loss-of-function (frameshift, stopgain or splicing) mutations, or *de novo* or likely-pathogenic (Phred-like CADD ≥25) missense mutations.
(PDF)

**S5 Table. Phenotypes observed for individual 16p12.1 homologs in *Drosophila* and *X. laevis* models.** Developmental and neuronal phenotypes observed with individual knockdown of 16p12.1 homologs in *Drosophila* and *X. laevis*.
(PDF)

**S6 Table. Sequences of oligonucleotides used for confirming gene knockdown in *Drosophila* and *X. laevis*.**
(PDF)

**S7 Table. *X. laevis* morpholino sequences used in this study.**
(PDF)

**S1 File. Stock list and genotypes of *Drosophila* lines used in the study.** This file details genotypes of tissue-specific drivers, and stock centers, stock numbers and genotypes of RNAi used

to knock down the four 16p12.1 homologs, as well as RNAi, mutant, or overexpression lines for homologs of "second-hit" genes and genes within conserved neurodevelopmental pathways. Genotypes of recombinant lines of 16p12.1 homologs crossed with interacting genes are also detailed, as well as the individual controls used for each experiment. Details of the number of homologs, fly lines, and crosses used for all interaction experiments are also provided. BDSC: Bloomington *Drosophila* Stock Center, VDRC: Vienna *Drosophila* Resource Center. (XLSX)

**S2 File. Differentially expressed genes and Gene Ontology enrichments for knockdown of 16p12.1 homologs.** List of differentially expressed genes, defined as log-fold change >1 or <-1 and false discovery rate (FDR) <0.05 (Benjamini-Hochberg correction), following RNA sequencing of *Drosophila* fly heads with *Elav-GAL4*-mediated knockdown of the four 16p12.1 homologs. Human homologs of differentially-expressed fly genes were identified using DIOPT [24]. Enriched Gene Ontology (GO) and PantherDB terms are also provided for differentially-expressed fly genes and their human homologs. (XLSX)

**S3 File. Summary of genetic interactions identified with 16p12.1 homologs in *Drosophila*.** This file details the genes that modulated the phenotypes of the 16p12.1 homologs, as well as the genetic interactions identified in this study. (XLSX)

**S4 File. Interaction scores and observed and expected values for all tested genetic interactions of 16p12.1 homologs in *Drosophila*.** (XLSX)

**S5 File. Connectors of 16p12.1 and 16p11.2 genes identified in a human brain network.** (XLSX)

**S6 File. Statistical analysis of experimental data.** This file provides details of all statistical information, including sample size, test statistics, p-values, and Benjamini-Hochberg FDR corrections for all data shown in the main and supplemental figures. Details for ANOVA tests include factors, degrees of freedom, tests statistics, and post-hot pariwise t-tests with Benjamini-Hochberg FDR corrections. (XLSX)

## Author Contributions

**Conceptualization:** Lucilla Pizzo, Santhosh Girirajan.

**Data curation:** Lucilla Pizzo, Micaela Lasser, Tanzeen Yusuff, Matthew Jensen, Vijay Kumar Pounraja.

**Formal analysis:** Lucilla Pizzo, Micaela Lasser, Matthew Jensen, Alexis T. Weiner, Vijay Kumar Pounraja, Arjun Krishnan, Santhosh Girirajan.

**Funding acquisition:** Melissa M. Rolls, Laura Anne Lowery, Santhosh Girirajan.

**Investigation:** Lucilla Pizzo, Micaela Lasser, Tanzeen Yusuff, Phoebe Ingraham, Emily Huber, Mayanglambam Dhruba Singh, Connor Monahan, Janani Iyer, Inshya Desai, Siddharth Karthikeyan, Dagny J. Gould, Sneha Yennawar, Alexis T. Weiner.

**Methodology:** Lucilla Pizzo, Santhosh Girirajan.

**Project administration:** Santhosh Girirajan.

**Resources:** Melissa M. Rolls, Laura Anne Lowery, Santhosh Girirajan.

**Software:** Matthew Jensen, Vijay Kumar Pounraja, Arjun Krishnan.

**Supervision:** Melissa M. Rolls, Laura Anne Lowery, Santhosh Girirajan.

**Validation:** Lucilla Pizzo, Tanzeen Yusuff, Phoebe Ingraham.

**Visualization:** Lucilla Pizzo.

**Writing – original draft:** Lucilla Pizzo, Micaela Lasser, Santhosh Girirajan.

**Writing – review and editing:** Lucilla Pizzo, Santhosh Girirajan.

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
