## [Decision Letter · Decision Letter 0]

5 Nov 2020

Dear Santhosh,

Thank you very much for submitting your Research Article entitled 'Functional assessment of the “two-hit” model for neurodevelopmental defects in Drosophila and X. laevis' to PLOS Genetics. Your manuscript was fully evaluated at the editorial level and by independent peer reviewers. The reviewers appreciated the attention to an important problem, but raised some substantial concerns about the current manuscript. Specifically, the reviewers had several questions regarding the presentation and interpretation of your findings. Based on the reviews, we will not be able to accept this version of the manuscript, but we would be willing to review again a much-revised version. We cannot, of course, promise publication at that time.

If you decide to revise the manuscript for further consideration at PLOS Genetics, please aim to resubmit within the next 60 days, unless it will take extra time to address the concerns of the reviewers, in which case we would appreciate an expected resubmission date by email to plosgenetics@plos.org.

We are sorry that we cannot be more positive about your manuscript at this stage. Please do not hesitate to contact us if you have any concerns or questions.

Yours sincerely,

Maja Bućan

Guest Editor

PLOS Genetics

Gregory Barsh

Editor-in-Chief

PLOS Genetics

Reviewer's Responses to Questions

**Comments to the Authors:**

Reviewer #1: This interesting and ambitious paper aims to assess interaction among genes of the 16p12.1 CNV with each other and with potential ‘second-hit’ candidate genes using drosophila and xenopus models. Although they present some intriguing data and hypotheses, substantial clarification could help the reader understand the scope, results, and interpretation of the study.

The manuscript includes terms like ‘interaction’, ‘epistasis’, ‘additive’, and ‘synergistic’ that need to be defined and then used precisely and supported, e.g. test whether the combined effects are significantly different than an additive prediction. Currently, the distinctions among these terms are unclear and may be misused (by a statistical definition), such as ‘additive interaction’. Other examples include results that polr3e and mosmo pairwise knockdown do not show more extreme phenotypes than one gene alone, yet this is referred to as ‘additive’, which typically means the sum of the effects of both. If one of the genes had no effect to begin with, the conclusion should just be no interaction.

In the abstract, the authors set up a comparison with the 16p11.2 CNV hypothesizing that its higher de novo frequency is associated with greater within-CNV interaction compared to 16p12.1 with greater inherited frequency and second-hit interaction. However, the authors do not present the 16p11.2 data for clear comparison in the results, so this major point in the abstract seems lost in the paper itself. Further, no context or negative control is given when reporting the number of interactions detected, e.g. are 18/55 and 29/61 unusual? What is the basis for comparison?

Only four out of seven genes in the region can be studied in these model organisms. How does that impact the conclusions made about interactions within the region, etc., when >40% of the genes are not assayed at all?

How do the authors interpret the severity of some single-gene partial knockdowns under their hypothesis of needing ‘second hits’? Two of the four genes tested show larval lethality with 40-60% expression, which is similar to human heterozygous levels. This doesn’t seem to square with being a good model to assess effects so modest they require interaction to have a phenotype in humans. Similarly, the authors report increased lifespan with one of the knockdowns and compare with other previously reported genes with no explanation of the meaning of this comparison or specific genes showing the same phenotype. Likewise, craniofacial defects are noted in the xenopus but not interpreted with respect to the human phenotype associated with this CNV.

When the results describe enhancement or reduction of single gene phenotypes in pairwise knockdowns, no interpretation of the direction is made with respect to a priori expectation. For example, based on whether human neurodevelopmental phenotypes are caused by loss of function vs. gain of function mutations, wouldn’t you predict a direction of interaction? Is it surprising to find roughly equal numbers of enhancers and suppressors? How were these known neurodevelopmental genes chosen if there was no expectation or prediction made? In general, it would be helpful to explain the specific hypothesis and how the genes were chosen to test it, e.g. neurodevelopmental or from transcriptome studies are a little vague to understand the study design. Is 12/24 enhancers in the ‘second hits’ different from 25/52 (if I’m counting this the same) other developmental genes tested?

In general, it gets a little confusing to count number of genes (in and out of the CNV) tested, number of total pairs (considering different lines from the same gene), number of unique pairs, etc., so greater clarity in how these are being counted, tested, and considered (and why) would be helpful.

Overall, this may be a valuable dataset and interesting theory, but clear delineation of the study design and interpretation and contextualization of the results is needed for the reader to understand it fully. It might be helpful to specify upfront whether some experiments are meant to be hypothesis generating but require additional testing or follow-up, e.g. specific pairs of genes that might interact in humans, and clarify hypotheses and testing approaches for the others, e.g. what were you intending to demonstrate about second-hit genes (vs neurodevelopmental genes?) and how did you specifically test that your results supported that hypothesis.

Reviewer #2: The paper entitled “Functional assessment of the “two-hit” model for neurodevelopmental defects in Drosophila and X. laevis” discusses the potential of the 16p12.1 deletion to sensitize probands to the advent of neurodevelopmental disease. In this manuscript, the authors suggest that the knockdown of 16p12.1 homologs can result in a diverse range of phenotypes, including delayed development, seizure susceptibility, brain malformations, and abnormal neuronal morphology. Two different pairwise assessments were performed examining the interaction of genes within the 16p12.1 deletion with each other and with putative “second-hit” genes identified from patients and examination of associated developmental pathways. In the cases studied, changes in eye phenotypes were observed in putatively interacting homologs from 11/15 families described. In support of their claims, the authors provide the following information:

1. The 16p12.1 deletion is inherited in more than 95% of individuals carrying the deletion, and affected children are more likely to carry another large CNV or deleterious mutation. The authors suggest this indicates that the 16p12.1 deletion confers significant risk for disease/sensitizes the genome, while other variation explains ultimate phenotype.

2. The loss of individual genes from the 16p11.2 and 3q29 regions is insufficient to explain the wide range of defects observed with the loss of the entire region.

3. Previous work with the 16p11.2 and 3q29 regions demonstrated pervasive interactions of homologs within regions associated with neurodevelopmental disease.

4. Four fly homologues (of the 7 human genes in 16p12.1) were identified: UQCR-C2, CG14182, Sin, and Cen. These have homologues in Xenopus as well.

5. Use of RNAi against UQCR-C2, CG14182, Sin, and Cen to induce either ubiquitous or tissue specific knockdown lead to a plethora of phenotypes, depending on the Gal4 line used. Knockdown efficiency of the RNAi was ~40-60%.

6. RNA-seq was performed on fly heads with pan neuronal knockdown of 16p21.1 homologues. Gene ontology analysis of differentially expressed genes demonstrated enrichments for cellular, developmental, and neuronal processes.

7. Xenopus was used to examine developmental phenotypes when homologues were knocked down using morpholinos. Two “strengths” of knockdown were used- partial knockdown (50%) and “further reduced” (approx. 10-40% of baseline). Knockdown resulted in alterations in craniofacial defects, axon outgrowth phenotypes, and decreased forebrain and midbrain size.

8. Use of GMR-Gal4 lines to study the eye revealed that knockdown of the 4 homologues could result in disruptions of the ommatidial organization, as identified by flynotyper. These phenotypes were more severe in the presence of Dicer2. Pairwise interactions (two RNAi lines) resulted in more severe phenotypes with some pairs.

9. Pairwise interactions in Xenopus was performed using partial (50%) knockdowns. Pariwise knockdown of mosmo and polr3e demonstrated reduced forebrain/midbrain size.

10. 166 pairwise gene interactions were attempted using flynotyper. 18 neurodevelopmental genes, 25 genes from the authors transcriptome studies, and 14 genes within functionally related pathways were identified as having an interactive effect.

11. 11. RNAi or knock outs of 24 homologs of patient specific “second hit” genes were tested with the RNAi of UQCR-C2, CG14182, Sin, and Cen. “Second hit” genes were shown to be able to act as both enhancers and suppressors, depending on the context. Similar interactions were observed in the Xenopus model, in particular the authors used the example of mosmo and set5, which had a significant effect on axon length.

After evaluating the documents provided, I believe this paper has several significant deficiencies that require further explanation or additional experiments to address. Due to the scope and nature of some of these critiques, it may not be possible to address them within a short period of time. The following should be addressed by the authors:

Major Critiques:

1. The authors note that RNAi knockdown of the 4 homologues resulted in a decrease in expression level of 40-60%. This calls into question if the remaining proteins are sufficient to restore functionality and if non-significant results may be due to residual functionality.

2. Figure 2A is confusing as it does not address the Gal4 line used to generate the reported phenotypes. Furthermore, pupal lethality is not reported anywhere in this paper save this graph.

3. The trajectory of the Elav-Gal4 control in figure 2B is hidden within the graph, and therefore impossible to evaluate.

4. Given that multiple controls are used throughout the paper, controls need to be labeled properly/completely throughout the figures, not just as control.

5. It is unclear how the authors chose which RNAi lines to display in Fig 2. From looking at the other figures, it is clear that multiple RNAi lines exist for each of the homologues, however only a small number of them are displayed here and in some cases are inconsistent between figures (such as CG14182GD2738_2). Furthermore, while the authors consider a trend to be “valid” when confirmed by multiple RNAi lines later in the paper, it is unclear what criterion is used to consider the lines corresponding to the homologues lines “valid”. For example: figure S8 when compared to S7. CG14182GD2738_2 showed a significant phenotype by itself when GMR-Gal4 was used to drive the RNAi. However, CG14182GD2738 was chosen for all future pair-wise experiments, and it is unclear why.

6. What the authors intend to indicate with lines 198-200 (uniquely disrupted GO terms) is not clear. GO terms are descriptive, and the authors should provide information about the genes affected, not a broad argument about systems affected.

7. 30 degrees C is a high temperature to raise flies at, and may accentuate phenotypes.

8. It is possible that the expression of any RNAi (or combination or RNAi’s) may be deleterious in the fly eye. The authors do not appear to have a control which can account for this. It would be optimal to show that an RNAi which does not have a potential target (RNAi against LacZ or similar) causes no effect.

9. As stated by the authors, “second-hit” genes were tested with multiple RNAi, mutant or overexpression lines. No information about the efficiency of knockdown on these RNAi lines is available.

10. “An interaction was considered to be validated when the observed trend was reproduced by multiple RNAi lines when available.” How validation was accomplished when multiple lines were unavailable is unclear.

11. “When epistatic effects were not confirmed with more than two RNAi lines, if available, the interaction was considered to be additive.” If epistatic effects cannot be confirmed with multiple lines, that should be noted and assigned a separate category.

**Have all data underlying the figures and results presented in the manuscript been provided?**

Reviewer #1: None

Reviewer #2: Yes

PLOS authors have the option to publish the peer review history of their article (what does this mean?). If published, this will include your full peer review and any attached files.

Reviewer #1: No

Reviewer #2: No

---

## [Decision Letter · Decision Letter 1]

15 Feb 2021

Dear Santhosh,

Thank you very much for submitting your Research Article entitled 'Functional assessment of the “two-hit” model for neurodevelopmental defects in Drosophila and X. laevis' to PLOS Genetics.

The manuscript was fully evaluated at the editorial level and by independent peer reviewers. The reviewers appreciated the attention to an important topic and we are positive about the work. However, we are asking you to clarify and explain points raised by the Reviewer 1.

 Your revisions should address the specific points regarding the interpretation of your findings on gene-gene interactions.

Yours sincerely,

Maja Bućan

Guest Editor

PLOS Genetics

Gregory Barsh

Editor-in-Chief

PLOS Genetics

Reviewer's Responses to Questions

**Comments to the Authors:**

Reviewer #1: Thank you for the detailed responses to reviewer concerns. I think the paper is improved, but the root of most of the concerns has not been addressed. It is also still extremely dense and difficult to parse, and the study design limitations continue to impact the overall conclusions and interpretation.

1) It’s very helpful to specifically test the multiplicative model, but still unclear in places the meaning of additive, multiplicative, and interaction. If you are testing both additive and multiplicative models for each pair and calling it interaction if both are rejected, that could be clarified. In your example of 120% and 140% - additive is 160% and multiplicative is 168%, so are you calling interactions those that are significantly below 160% or above 168%? In the response you indicate that multiplicative is the best model of ‘interaction’ for functional effects, but in your example, you consider interaction to be deviation from the multiplicative model, so how you are using these could be further clarified.

2) The big-picture interpretation of your data as 16p12.1 genes being less likely to interact with each other and more likely to interact with second-hits is still difficult to support with the comparisons you have made. For example, the 16p11.2 region has more genes tested for interactions, so you might randomly sample 4 at a time and then compare 16p12.1 with the median of that set. Even more problematic is the ‘second-hit’ hypothesis expanded to candidate gene sets and no control sets. If you have selected genes by those most likely to interact because of their developmental role, etc., you need a matched comparison set to really test a specific hypothesis, e.g. are developmental genes more likely to interact than matched brain-expressed genes? are the second-hits observed in families more likely to interact than other genes that would be classified as ‘pathogenic’ in clinical testing for developmental disorders? What is the comparison for your selected gene lists? How do you explain the attenuation effect if your hypothesis is that the 16p12.1 genes sensitize the genome to developmental effects? The majority of your selected genes seem to enhance, but patient 2nd hits seem to suppress- how do you interpret that with respect to your hypotheses?

3) I still have trouble sorting out well-rationalized a priori hypotheses from post hoc interpretation of observations. I appreciate the intention of the table delineating hypotheses, but these are still pretty vague to me. If your hypothesis is “16p12.1 homologs interact towards neurodevelopmental phenotypes,” does 3/12 support this? Is that more than non-CNV sets of 4 consecutive genes? You identified at least one interaction, but a smaller proportion were interacting than for 16p11.2, so a more specific hypothesis and comparison to a ‘control’ or ‘alternative’ condition would be helpful in clarifying the hypothesis.

Reviewer #2: The authors have thoroughly and completely addressed the concerns raised in peer review in this more comprehensive version.

**Have all data underlying the figures and results presented in the manuscript been provided?**

Reviewer #1: None

Reviewer #2: Yes

PLOS authors have the option to publish the peer review history of their article (what does this mean?). If published, this will include your full peer review and any attached files.

Reviewer #1: No

Reviewer #2: No

---

## [Editor Report · Decision Letter 2]

16 Mar 2021

Dear Santhosh,

We are pleased to inform you that your manuscript entitled "Functional assessment of the “two-hit” model for neurodevelopmental defects in Drosophila and X. laevis" has been editorially accepted for publication in PLOS Genetics. Congratulations!

Yours sincerely,

Maja Bućan

Guest Editor

PLOS Genetics

Gregory Barsh

Editor-in-Chief

PLOS Genetics

Comments from the reviewers (if applicable):

**Data Deposition**

http://datadryad.org/submit?journalID=pgenetics&manu=PGENETICS-D-20-01394R2

**Press Queries**

---

## [Editor Report · Acceptance letter]

31 Mar 2021

PGENETICS-D-20-01394R2 

Functional assessment of the “two-hit” model for neurodevelopmental defects in Drosophila and X. laevis 

Dear Dr Girirajan, 

We are pleased to inform you that your manuscript entitled "Functional assessment of the “two-hit” model for neurodevelopmental defects in Drosophila and X. laevis" has been formally accepted for publication in PLOS Genetics! Your manuscript is now with our production department and you will be notified of the publication date in due course.

With kind regards,

Katalin Szabo

PLOS Genetics

On behalf of:
